# DOMINO: a network-based active module identification algorithm with reduced rate of false calls

Hagai Levi[1], Ran Elkon[2,3,†] & Ron Shamir[1,*,†]

## Abstract

Algorithms for active module identification (AMI) are central to analysis of omics data. Such algorithms receive a gene network and nodes' activity scores as input and report subnetworks that show significant over-representation of accrued activity signal ("active modules"), thus representing biological processes that presumably play key roles in the analyzed conditions. Here, we systematically evaluated six popular AMI methods on gene expression and GWAS data. We observed that GO terms enriched in modules detected on the real data were often also enriched on modules found on randomly permuted data. This indicated that AMI methods frequently report modules that are not specific to the biological context measured by the analyzed omics dataset. To tackle this bias, we designed a permutation-based method that empirically evaluates GO terms reported by AMI methods. We used the method to fashion five novel AMI performance criteria. Last, we developed DOMINO, a novel AMI algorithm, that outperformed the other six algorithms in extensive testing on GE and GWAS data. Software is available at https://github.com/Shamir-Lab.

**Keywords** biological networks; enrichment analysis; GO terms; module discovery; omics

**Subject Category** Computational Biology

**Mol Syst Biol. (2021) 17: e9593**

## Introduction

The maturation of high-throughput technologies has led to an unprecedented abundance of omics studies. With the ever-increasing volume of publicly available genomic, transcriptomic, and proteomic data (Perez-Riverol *et al*, 2019), it remains a challenge to uncover biological and biomedical insights out of it. As data accumulated over the last two decades strongly indicate that the functional organization of the cell is fundamentally modular, a leading approach to this challenge relies on biological networks, simplified yet solid mathematical abstractions of complex intra-cellular systems. In these networks, each node represents a cellular subunit (e.g., a gene or its protein product) and each edge represents a relationship between two subunits (e.g., a physical interaction between two proteins) (reviewed in (McGillivray *et al*, 2018)). A biological module is described as a connected subnetwork of—molecules that take part in a common biological process. As such, modules are regarded as functional building blocks of the cell (Hartwell *et al*, 1999; Alon, 2003; Barabási & Oltvai, 2004).

The challenge of identifying modules in biological networks, frequently referred to as *network-based module identification* or *community detection*, has yielded many computational methods (for a recent comparative study see (Choobdar *et al*, 2019)), and successfully identified molecular machineries that perform basic biological functions and underlie pathological phenotypes (Ideker & Sharan, 2008; Barabási *et al*, 2011). However, such analysis is limited as it is based on a static snapshot of an abstract universal cell provided by the network, while the state of the cell greatly varies under different physiological conditions. One very powerful way to overcome this limitation is by integrating the analysis of omics data and biological networks. This approach overlays molecular profiles (e.g., transcriptomic, genomic, proteomic, or epigenomic profiles) on the network, by scoring nodes or weighting edges. This additional layer of condition-specific information is then used to detect modules that are relevant to the analyzed molecular profile (Mitra *et al*, 2013). A prominent class of such algorithms seek subnetworks that show a marked over-representation of accrued node scores (Ideker *et al*, 2002; Mitra *et al*, 2013; preprint: Reyna *et al*, 2020). Modules detected by such methods are often called "*active modules*," and following this terminology we refer to nodes' scores as "*activity scores*" and to the task of detecting active modules using such scores as *Active Module Identification* (AMI). (The task is sometimes called community detection with node attributes (Yang *et al*, 2014)). Hereafter, for brevity, where clear from the context, we refer to active modules reported by AMI methods simply as modules.

Modules detected by AMI algorithms are expected to capture context-specific molecular processes that correlate with the specific

1   The Blavatnik School of Computer Science, Tel Aviv University, Tel Aviv, Israel
2   Department of Human Molecular Genetics and Biochemistry, Sackler School of Medicine, Tel Aviv University, Tel Aviv, Israel
3   Sagol School of Neuroscience, Tel Aviv University, Tel Aviv, Israel
    *Corresponding author. Tel: +972 36405383; E-mail: rshamir@tau.ac.il
    † These authors contributed equally to this work

cellular state or phenotype that is probed by the analyzed omics profile (Mitra *et al*, 2013). Different AMI methods use different scoring metrics, objective functions, and constraints. For example, activity scores may be binary or continuous, the objective function could penalize for including low-scoring nodes, and constraints can limit the number of "non-active" nodes in a module. While the metrics by which modules are scored may differ from one method to another, the activity scores are always derived from the data (e.g., $log_2$(fold–change of expression) for transcriptomic data). As the AMI problem has been proven to be NP-hard (Ideker *et al*, 2002), many heuristics were suggested for solving it (Mitra *et al*, 2013; Creixell *et al*, 2015).

Solutions reported by AMI methods comprise a set of active modules. A common downstream analysis is to ascribe each module some biological annotations that will point to the biological processes that it affects (Cerami *et al*, 2010; Leiserson *et al*, 2015; Barel & Herwig, 2018). This is most commonly done by testing enrichment of the modules for GO terms (The Gene Ontology Consortium, 2019). AMI solutions would ideally break down complex biological states into distinct functional modules, each mediating one or several highly related biological processes. For example, biological responses to genotoxic stress often comprise the concurrent activation and repression of multiple biological processes (e.g., DNA repair, cell-cycle arrest, apoptosis), each mediated by a single or a few dedicated signaling pathways (Ashcroft *et al*, 2000; Kyriakis & Avruch, 2012).

Another key advantage of AMI methods is the amplification of weak signals, where a reported active module comprises multiple nodes that individually have only marginal scores, but when considered in aggregate score significantly higher. This merit of AMI methods is especially critical for the functional interpretation of Genome-Wide Association Studies (GWASs) (Carter *et al*, 2013; Cowen *et al*, 2017). Numerous GWASs conducted over the last decade have demonstrated that the genetic component of complex diseases is highly polygenic (Khera *et al*, 2018; Musunuru & Kathiresan, 2019; Sullivan & Geschwind, 2019), affected by hundreds or thousands of genetic variants, the vast majority of which have only a very subtle effect. Therefore, most "risk SNPs" do not pass statistical significance when tested individually after correcting for multiple testing (Stringer *et al*, 2011; Boyle *et al*, 2017). This stresses the need for computational methods that consider multiple genetic elements together, to allow detection of biological pathways that carry high association signal. As a first step in this challenge, gene-level scores are inferred from the scores of the genetic variants that map to the same gene (de Leeuw *et al*, 2015; Lamparter *et al*, 2016). These gene scores then serve as activity scores by AMI methods for integrated analysis of GWAS data and biological networks. Recently, such analyses successfully elucidated novel process that are implicated in the pathogenesis of inflammatory bowel disease, Schizophrenia, and Type-2 diabetes (Chang *et al*, 2015; Nakka *et al*, 2016; Fernández-Tajes *et al*, 2019).

In this study, we first aimed to systematically evaluate popular AMI algorithms across multiple gene expression (GE) and GWAS datasets based on enrichment of the called modules for GO terms. Remarkably, our analysis revealed that AMI algorithms often reported modules that showed enrichment for a high number of GO terms even when run on permuted datasets. Moreover, some of the GO terms that were often enriched on permuted datasets were also enriched on the original dataset, indicating that AMI solutions frequently include modules that are not specific to the biological context measured by the analyzed omics dataset. To tackle this bias, we designed a procedure for validating the functional analysis of AMI solutions by comparing them to null distributions obtained on permuted datasets. We used the empirically validated set of GO terms to define novel metrics for evaluation of AMI algorithm results. Finally, we developed DOMINO (Discovery of active Modules In Networks using Omics)—a novel AMI method and showed its advantage in comparison it to the previously developed algorithms.

## Results

### AMI algorithms suffer from a high rate of non-specific GO term enrichments

We set out to evaluate the performance of leading AMI algorithms. Our analysis included six algorithms—jActiveModules (Ideker *et al*, 2002) in two strategies: greedy and simulated annealing (abbreviated jAM_greedy and jAM_SA, respectively), BioNet (Beisser *et al*, 2010), HotNet2 (Leiserson *et al*, 2015), NetBox (Cerami *et al*, 2010), and KeyPathwayMiner (Baumbach *et al*, 2012) (abbreviated KPM). These algorithms were chosen based on their popularity, computational methodology, and diversity of original application (e.g., gene expression data, somatic mutations) (Appendix Table S1). As we wished to test these algorithms extensively, we focused on those that had a working tool/codebase that can be executed in a stand-alone manner, have reasonable runtime, and could be applied to different omics data types. Details on the execution procedure of each algorithm are available in the Appendix. We applied these algorithms to two types of data: (1) a set of ten gene expression (GE) datasets of diverse biological physiologies (Appendix Table S2) where gene activity scores correspond to differential expression between test and control conditions, and (2) a set of ten GWAS datasets of diverse pathological conditions (Appendix Table S3) where gene activity scores correspond to genetic association with the trait (Methods). Note that for uniformity, we use the term activity also for the GWAS scores. In our analyses, we mainly used the Database of Interacting Proteins (DIP; (Xenarios *et al*, 2002)) as the underlying global network. Although the DIP network is relatively small—comprising about 3000 nodes and 5000 edges, in a recent benchmark analysis (Huang *et al*, 2018), it got the best normalized score on recovering literature-curated disease gene sets, making it ideal for multiple systematic executions.

First, applying the algorithms to the GE and GWAS datasets we observed that their solutions showed high variability in the number and size of active modules they detected (Appendix Fig S1 and Appendix Fig S2). On the GE datasets, jAM_SA tended to report a small number of very large modules while HotNet2 usually reported a high number of small modules (Appendix Fig S1). jAM_SA showed the same tendency for reporting large modules also on the GWAS datasets (Appendix Fig S2). Next, to functionally characterize the solutions obtained by the algorithms, we tested the modules for enriched GO terms using the hypergeometric (HG) test with the genes in the entire network as the background set. Specifically, we used GO terms from the Biological Process (BP) ontology, using

only terms with 5-500 genes. To avoid potential bias caused by the underlying network and datasets, we excluded from each GO class genes that were included in it based on physical interaction, expression pattern, genetic interaction, or mutant phenotype (GO evidence codes: IPI, IEP, IGI, IMP, HMP, HGI, and HEP). Next, as part of our evaluation analysis, we applied the algorithms also on random datasets generated by permuting the original gene activity scores uniformly at random. Notably, we observed that modules detected on the permuted datasets, too, were frequently enriched for GO terms (Fig 1A) Moreover, different algorithms showed varying degree of overlap between the enriched terms obtained on real and permuted datasets (Fig 1B). These findings imply that many terms reported by AMI algorithms do not stem from the specific biological condition that was assayed in each dataset, but rather from other non-specific factors that bias the solution, such as the structure of the network, the methodology of the algorithm, and the distribution of the activity scores.

**A permutation-based method for filtering false GO terms**

The high overlap between sets of enriched GO terms obtained on real and permuted datasets indicates that the results of most AMI algorithms tested are highly susceptible to false calls that might lead to functional misinterpretation of the analyzed omics data. We looked for a way to filter out such non-specific terms while preserving the ones that are biologically meaningful in the context of the analyzed dataset. For this purpose, we developed a procedure called the EMpirical Pipeline (EMP). It works as follows: Given an AMI algorithm and a dataset, EMP permutes genes' activity scores in the dataset and executes the algorithm. For each module reported by the algorithm, it performs GO enrichment analysis. The overall reported enrichment score for each GO term is its maximal score over all the solution's modules (Fig 2A). The process is repeated many times (typically, in our analysis, 5,000 times), generating a background distribution per GO term (Fig 2B). Next, the algorithm and the enrichment analysis are run on the real (i.e., non-permuted) dataset (Fig 2C). Denoting the background CDF obtained for GO term $t$ by $F_t$, the empirical significance of $t$ with enrichment score s is $e(t) = 1 - F_t(s)$. EMP reports only terms $t$ that passed the HG test (q-value $\leq 0.05$ on the original data) and had empirical significance $e(t) \leq 0.05$ (Fig 2D). We call such terms *empirically validated GO terms (EV terms)*. In addition, for each AMI algorithm solution, we define the *Empirical-to-Hypergeometric Ratio* (EHR) as the fraction of EV terms out of the GO terms that passed the HG test (Fig 2E and F).

**The DOMINO algorithm**

Our results demonstrated that popular AMI algorithms often suffer from high rates of false GO terms. While the EMP method is a potent way for filtering out non-specific GO term calls from AMI solutions, this procedure is computationally demanding, as it requires several thousands of permutation runs. In our analyses, using a 44-cores server, EMP runs typically took several days to complete, depending on the algorithm and the dataset. Seeking a more frugal alternative that can be used on a desktop computer, we developed a novel AMI algorithm called DOMINO (*Discovery of active Modules In Networks using Omics*), with the goal of producing highly confident active

modules characterized by high validation rate (that is, high EHR values).

DOMINO receives as input a set of genes flagged as the *active genes* in a dataset (e.g., the set of genes that in the analyzed transcriptomic dataset passed a test for differential expression) and a network of gene interactions, aiming to find disjoint connected subnetworks in which the active genes are over-represented. DOMINO has four main steps:

0. Partition the network into disjoint, highly connected subnetworks (*slices*).
1. Detect *relevant slices* where active genes are over-represented
2. For each relevant slice S
   a. Refine S to a sub-slice S'
   b. Repartition S' into putative modules
3. Report as final modules those that are over-represented by active genes.

***Step 0—Partitioning the network into slices***
This time-consuming preprocessing step is done once per network (and reused for any analyzed dataset). In this step, the network is split into disjoint subnetworks called slices. Splitting is done using a variant of the Louvain modularity algorithm (Blondel *et al*, 2008) (Methods). Each connected component in the final network that has more than three nodes is defined as a *slice* (Fig 3A).

***Step 1—Detecting relevant slices***
Each slice that contains more active nodes than a certain threshold (see Methods) is tested for active nodes over-representation using the hypergeometric (HG) test, correcting the *P*-values for multiple testing using FDR (Benjamini & Hochberg, 1995). In this initial step, we use a lenient threshold of q-values < 0.3 to accept a slice as a *relevant* one (Fig 3B).

***Step 2a—Refining the relevant slices into sub-slices***
From each slice, the algorithm extracts a single connected component that captures most of the activity signal. The single component is obtained by solving the Prize Collecting Steiner Tree (PCST) problem (Johnson *et al*, 2000) (Methods). The resulting subgraph is called a *sub-slice* (Fig 3C).

***Step 2b—Partitioning sub-slices into putative active modules***
Each sub-slice that is not over-represented by active nodes and has more than 10 nodes is partitioned using the Newman–Girvan algorithm (Methods). The resulting parts, as well as all the sub-slices from step 2a of $\leq 10$ nodes, are called *putative active modules* (Fig 3D).

***Step 3—Identifying the final set of active modules***
Each putative active module is tested for over-representation of active nodes using the HG test. In this step, we correct for multiple testing using the more stringent Bonferroni correction. Those with q-value < 0.05 are reported as the final active modules (Fig 3E).

**Systematic evaluation of AMI algorithms on gene expression and GWAS datasets**

We next carried out a comparative evaluation of DOMINO, and the six AMI algorithms described above over the same ten GE and ten

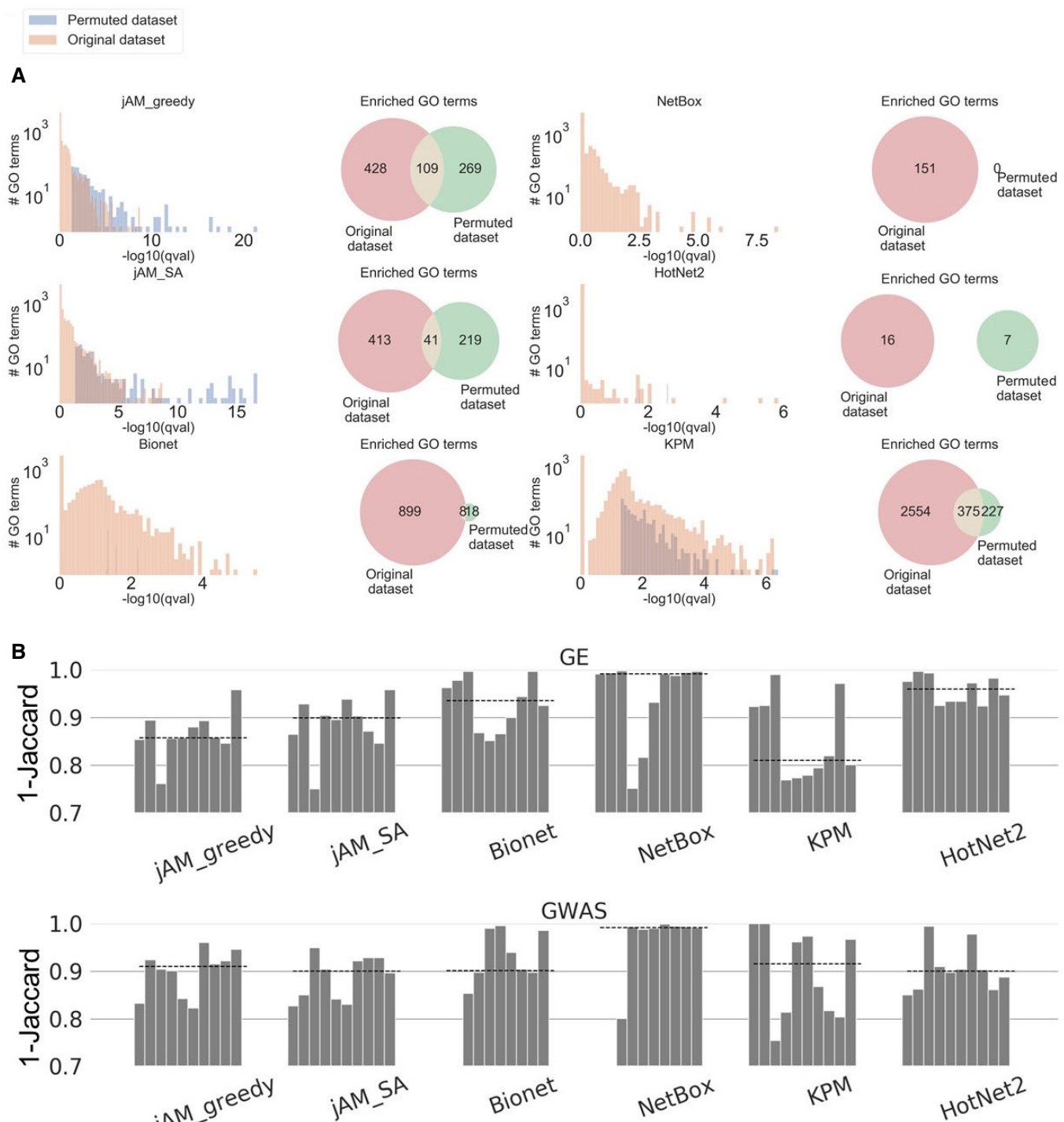

**Figure 1. Comparison between GO enrichment results derived by AMI algorithms from original and permuted activity scores.**

A  Comparison of GO enrichment results obtained on the original CBX GE dataset and on one random permutation of the original gene activity scores of this dataset. The histograms show the distributions of GO enrichment scores obtained for the modules detected on the original and permuted datasets. The Venn diagrams show the overlap between the GO terms detected in the two solutions.

B  Comparison of GO terms reported on the original and permuted datasets. We used 1-Jaccard score to measure the dissimilarity between the GO terms detected on the two datasets. Values closer to 1 indicate low similarity (that is, lower bias). Each bar shows, per algorithm, this measure on the ten datasets, averaged over 100 random permutations. Datasets are ordered from left to right as in Appendix Tables S2 and S3. Dashed lines show the median score.

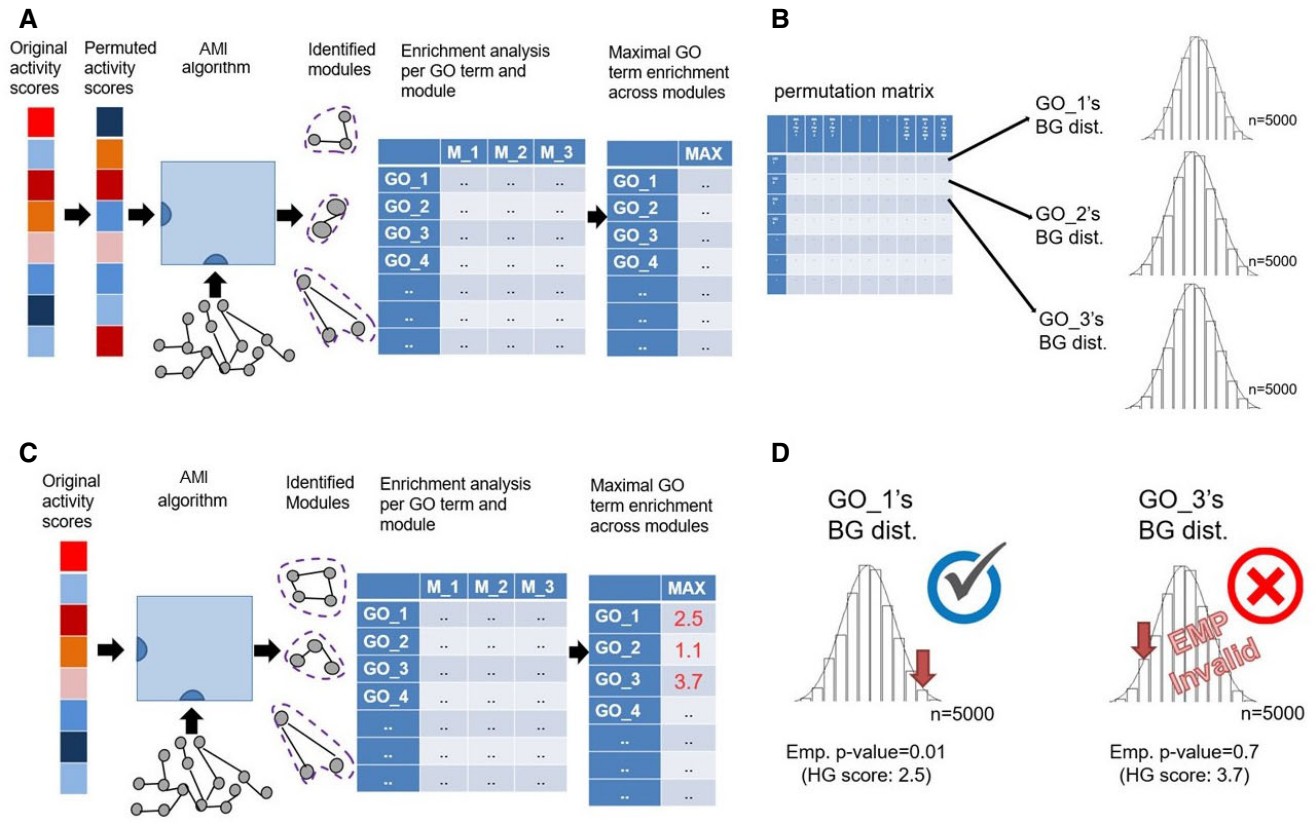

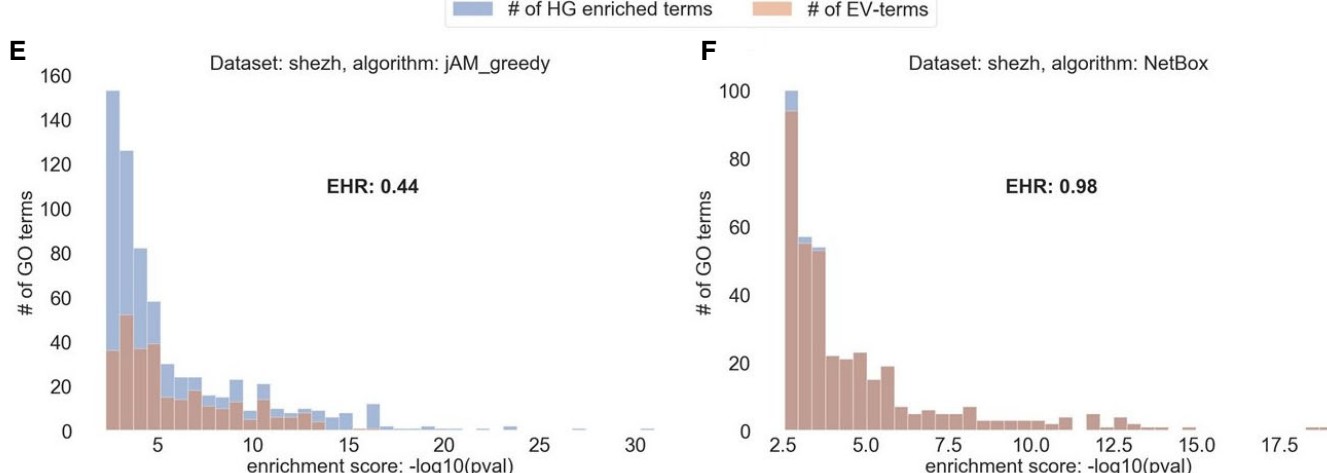

**Figure 2. Overview of the EMpirical Pipeline (EMP) procedure.**

A    The AMI algorithm and the GO enrichment analysis are applied on multiple (typically, $n = 5{,}000$) permuted activity scores.

B    A null distribution of enrichment scores (-log10(pval)) is produced per GO term.

C    The AMI algorithm is applied to the original (un-permuted) activity scores, to calculate the real GO enrichment scores.

D    For each GO term, the real enrichment score is compared to its corresponding empirical null distribution to derive an empirical score. In this example, GO_3 passed the HG test, but failed the empirical test and thus was filtered out.

E, F    Distributions of HG enrichment scores for all the GO terms that passed the HG test and for the subset of the EV terms obtained on the SHEZH GE dataset by jActiveModules with greedy strategy (E) and NetBox (F). EHR measures the ratio between the number of EV terms and the number of GO terms that passed the HG test. The high EHR obtained by NetBox (close to 1.0) demonstrates the advantage of this algorithm in avoiding false terms.

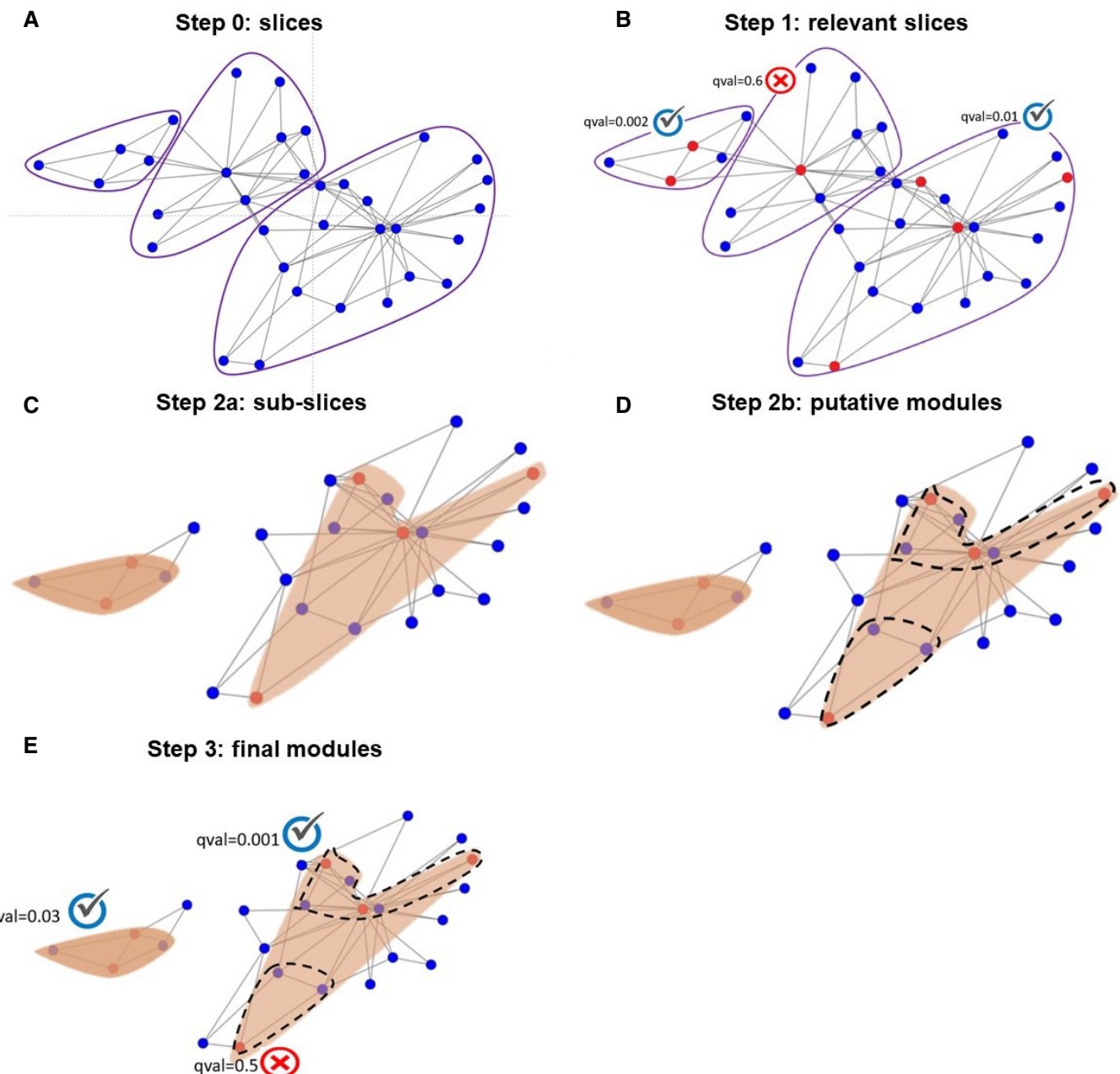

**Figure 3.  Schematic illustration of DOMINO.**

A   The global network is partitioned by the Louvain modularity algorithm into slices (encompassed in purple line).
B   A slice is considered relevant if it passes a moderate HG test for enrichment for active nodes (FDR q ≤ 0.3).
C   For each relevant slice the most active sub-slice is identified using PCST (red areas).
D   Sub-slices are further partitioned into putative active modules using the Newman–Girvan (NG) modularity algorithm.
E   Each putative active module that passes a strict over-representation test for active nodes (Bonferroni qval ≤ 0.05) is included in the final solution.

GWAS datasets. This evaluation task is challenging as there are no "gold-standard" solutions to benchmark against. To address this difficulty, we introduce five novel scores as evaluation criteria of AMI algorithms. These scores are based on the EMP method and the GO terms that pass this empirical validation procedure. The scores are described in Methods, and the results on all algorithms are summarized in Figs 4–6.

**EHR (Empirical-to-Hypergeometric Ratio)**
EHR summarizes the tendency of an AMI algorithm to capture biological signals that are specific to the analyzed omics dataset, i.e., GO terms that are enriched in modules found on the real but not on permuted data. EHR has values between 0 and 1, with higher values indicating better performance. In our evaluation, DOMINO and NetBox scored highest on EHR. In both GE and GWAS datasets, DOMINO performed

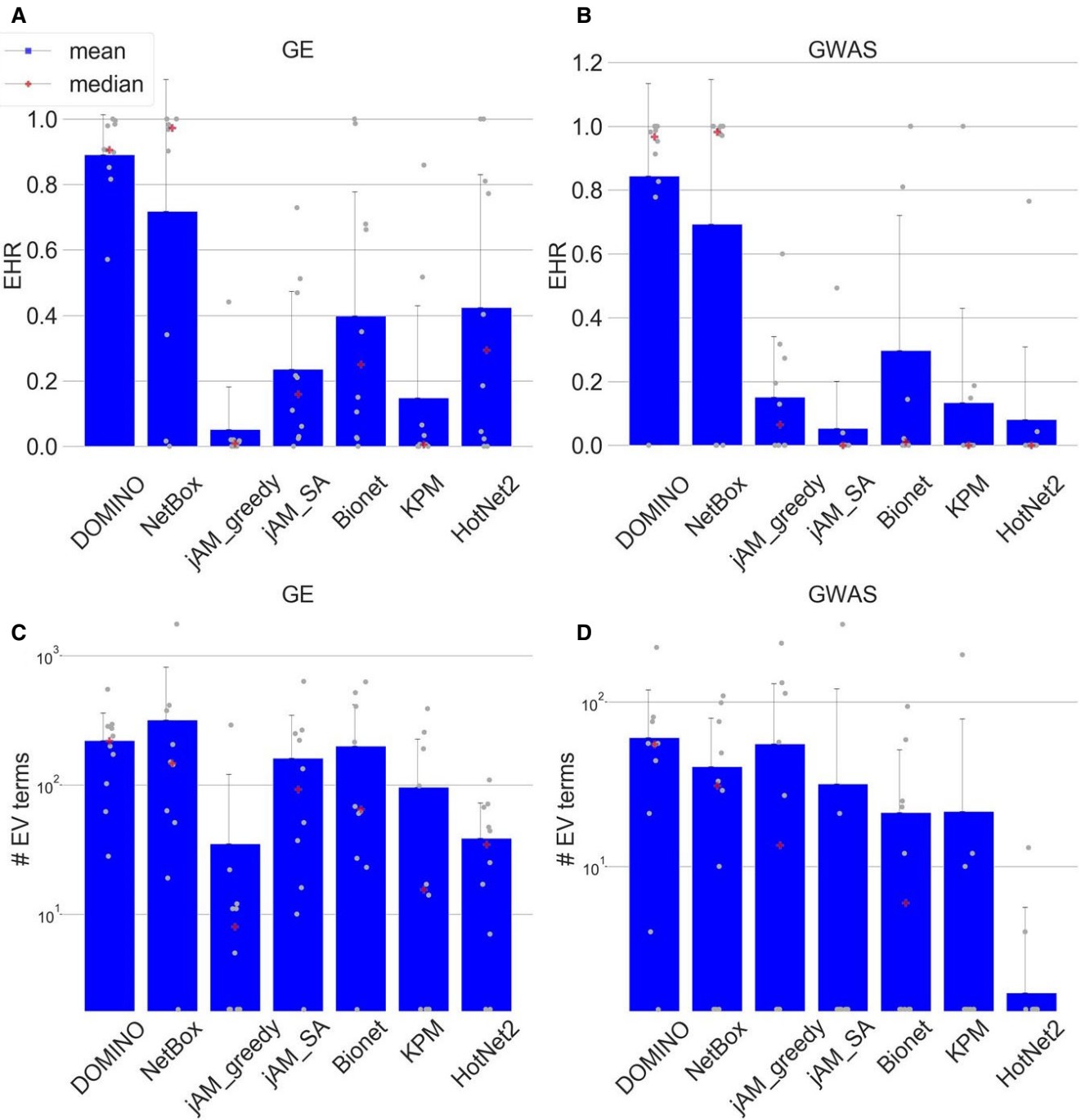

**Figure 4.  EHR and number of reported terms.**

A   Average EHR for the GE datasets.
B   Average EHR for the GWAS datasets.
C   The number of EV terms reported for the GE datasets.
D   The number of EV terms reported for the GWAS datasets.

Data information: The gray dots indicate results for each dataset (*n* = 10, each representing a different biological condition). Error bars indicate SD across datasets.

best with an average above 0.8. (Fig 4A and B). Importantly, these high EHR levels were not a result of reporting low number of terms: DOMINO reported on average more enriched GO terms than the other algorithms, except NetBox on GE datasets (Fig 4C and D).

**Module-level EHR (mEHR)**

While the EHR characterizes a solution as a whole by considering the union of GO terms enriched on any module, biological insights are often obtained by functionally characterizing each module

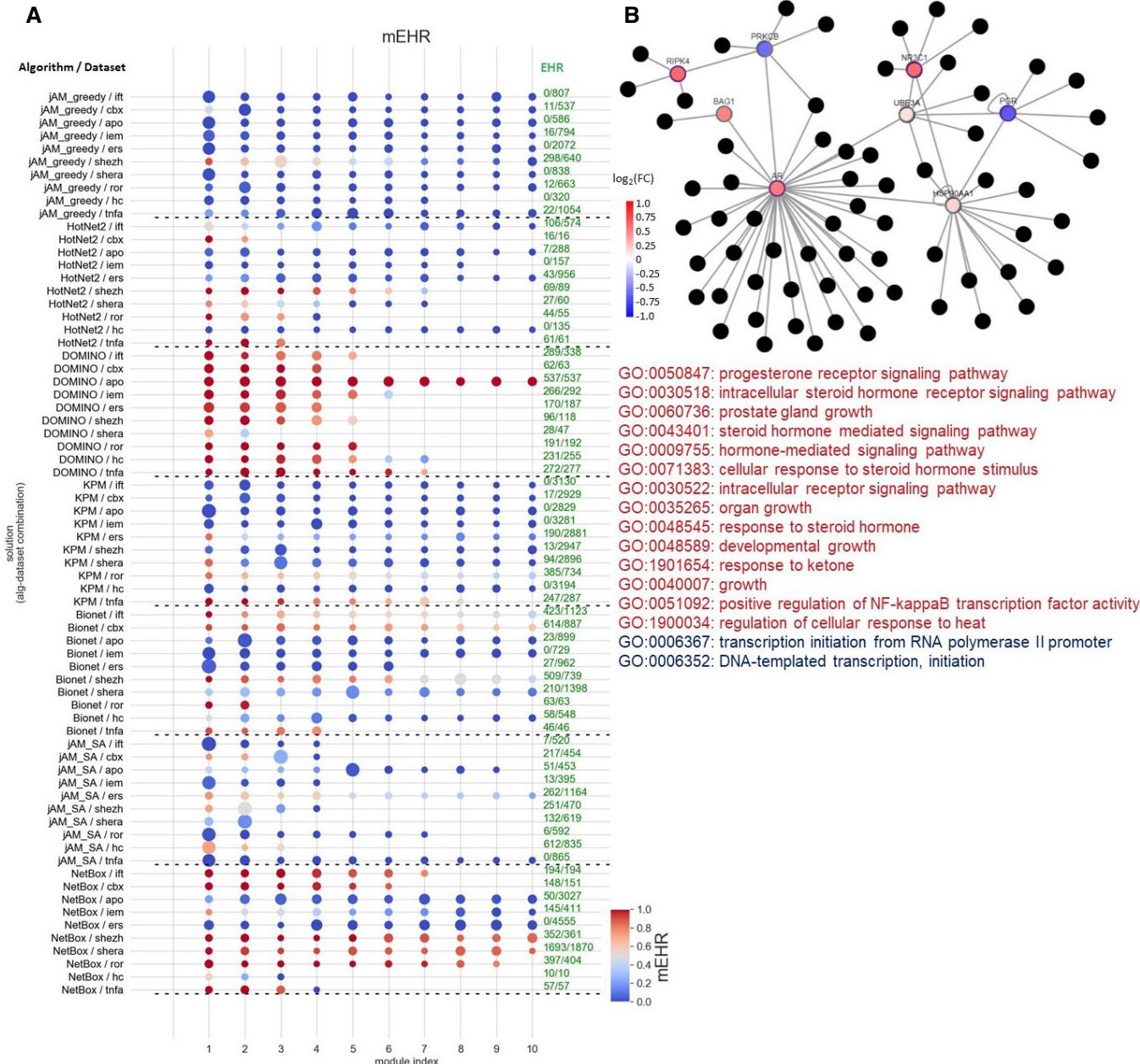

**Figure 5. AMI algorithms evaluated by the module-level EHR (mEHR) criterion on GE datasets.**

A  mEHR scores for each algorithm and dataset. Up to ten top modules are shown per dataset, ranked by their mEHR. Dot size represents module's size. The EHR column in green shows the number of EV terms and the number of significant terms found.

B  An example of a module from the solution reported by NetBox on the ROR dataset (mEHR = 0.88). The nodes' color indicates expression fold change (log scale) in the dataset. The black nodes are the network neighbors of the module's nodes. Nodes with purple border have significant activity scores (that is, significant differential expression; qval < 0.05). The EV terms for this module are shown in red and those that did not pass the empirical validation in blue.

individually. We therefore next evaluated the EHR of each module separately. Specifically, for each module, we calculated the fraction of its EV terms out of the HG terms detected on it (Methods). Modules with high mEHR score are the biologically most relevant ones, in the context of the analyzed omics dataset, while modules with low mEHR mostly capture non-specific signals. The comparison between mEHR scores obtained by the different AMI algorithms

is summarized in Fig 5A. Notably, solutions can have a broad range of mEHR scores (for example, in NetBox solution on the IEM dataset, the best module has mEHR = 0.78 while the poorest has mEHR = 0). To summarize the results over multiple modules, we averaged the k top scoring modules (from k = 1 to 20; Fig 6A). In this criterion, DOMINO scored highest, followed by NetBox. The results for GWAS datasets are shown in Figs EV1 and EV2A.

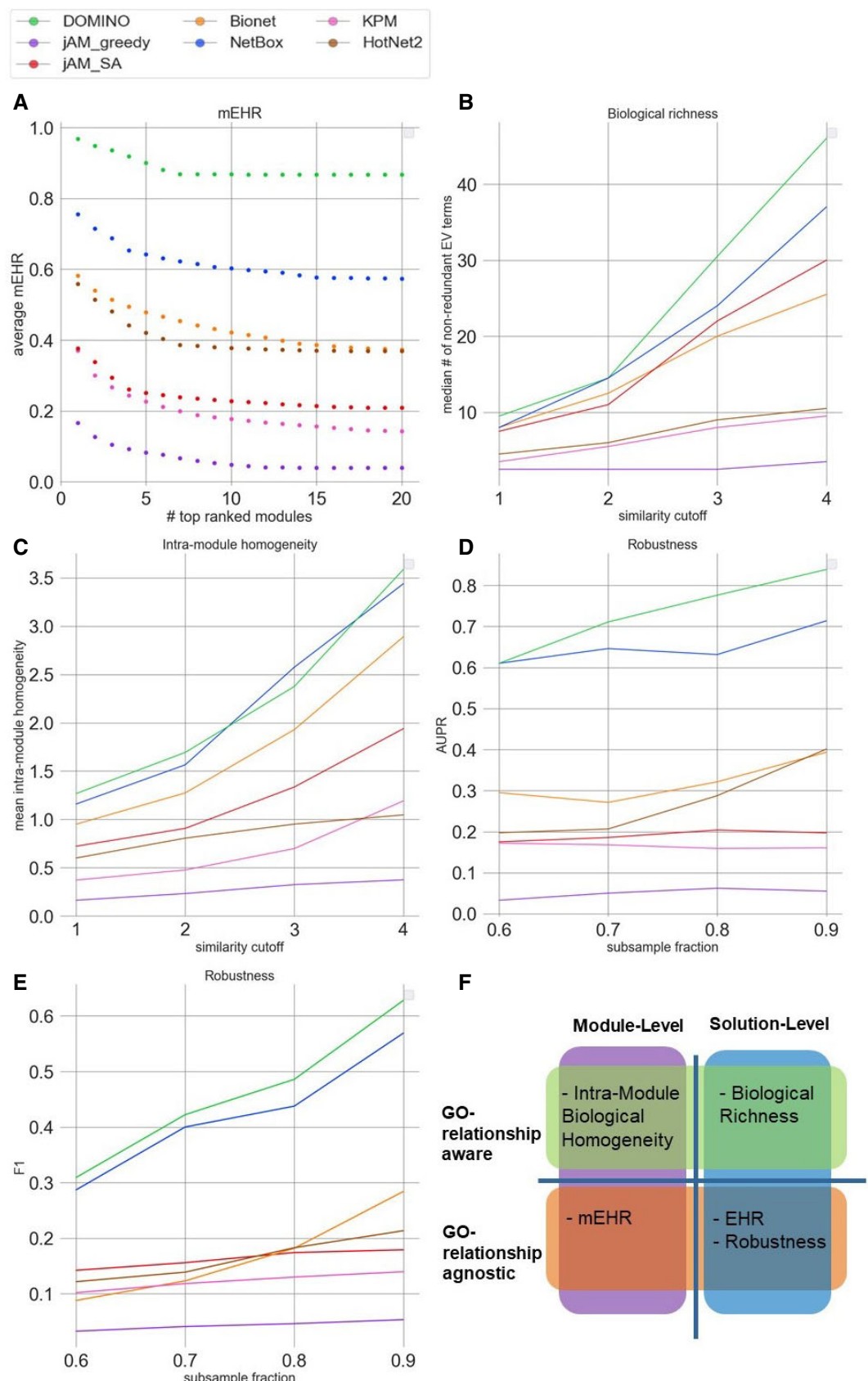

**Figure 6.**

**Figure 6. Evaluation results for the GE datasets.**

A  Module-level EHR scores. The plot shows the average mEHR score of the k top modules, as a function of k in each solution. Modules were ranked, for each solution, by their mEHR score. Then, for each solution with n modules we calculate the average mEHR of the top min(k, n) modules. Finally, we averaged the results and got the average mEHR of an algorithm.

B  Biological richness. The plot shows the median number of non-redundant terms (richness score) as a function of the Resnik similarity cutoff (Methods).

C  Intra-module homogeneity scores as a function of the similarity cutoff.

D  Robustness measured by the average AUPR over the datasets, shown as a function of the subsampling fraction.

E  Robustness measured by the average F1 over the datasets shown as a function of the subsample fraction. In (D, E), 100 samples were drawn and averaged for each dataset and subsampling fraction.

F  A breakdown of the evaluation criteria by their properties. Richness, EHR, and robustness score solutions based only on the whole set of the reported GO terms, without taking into account the results for individual modules. In contrast, mEHR and intra-module homogeneity score solutions in a module-aware fashion. From another perspective, biological richness and intra-module homogeneity consider the relations among the reported GO terms, while EHR, mEHR, and robustness do not.

Furthermore, the EMP procedure enhances the functional interpretation of each module by distinguishing between its enriched GO terms that are specific to the real data (i.e., the EV terms) and those that are recurrently enriched also on permuted ones. This utility of EMP is demonstrated, as one example, on a module detected by NetBox on the ROR GE dataset (Fig 5B). This study examined roles of the ROR2 receptor in breast cancer progression, and the GO terms that passed EMP validation are highly relevant for this process (e.g., GO terms related to steroid hormone-mediated signaling pathways). In contrast, GO terms that failed passing this validation procedure represent less specific processes (e.g., "DNA-templated transcription, initiation").

### Biological richness

This criterion aims to measure the diversity of biological processes captured by a solution. Our underlying assumption here is that biological systems are complex and their responses to triggers typically involve the concurrent modulation of multiple biological processes. For example, genotoxic stress concurrently activates DNA damage repair mechanisms and apoptotic pathways and suppresses cell-cycle progression. However, merely counting the number of EV terms of a solution would not faithfully reflect its biological richness because of the high redundancy between GO terms. This redundancy stems from overlaps between sets of genes assigned to different GO terms, mainly due to the hierarchical structure of the ontology. We therefore used REVIGO (Supek et al, 2011) to derive a non-redundant set of GO terms based on semantic similarity scores (Resnik, 1999; Lord et al, 2003). We defined the biological richness score of a solution as the number of its non-redundant EV terms (Methods). The results in Fig 6B show that on the GE datasets, DOMINO, and NetBox performed best. On the GWAS datasets, DOMINO performed best (Fig EV2B). Note that the interpretation of this criterion is condition dependent: High biological richness can be revealing or an indication of spurious results.

### Intra-module homogeneity

While high biological diversity (richness) is desirable at the solution level, each individual module should ideally capture only a few related biological processes. Solutions in which the entire response is partitioned into separate modules where each represents a distinct biological process are easier to interpret biologically and are preferred over solutions with larger modules that represent several composite processes. To reflect this preference, we introduced the intra-module homogeneity score, which quantifies how functionally homogeneous the EV terms captured by each module are (Methods;

Appendix Fig S3). For each solution, we take the average score of its modules. On the GE datasets, NetBox performed best (Fig 6C). On the GWAS datasets, DOMINO scored highest (Fig EV2C).

### Robustness

This criterion measures how robust an algorithm's results are to subsampling of the data. It compares the EV terms obtained on the original dataset with those obtained on randomly subsampled datasets. Running 100 subsampling iterations and using the EV terms found on the original dataset as the gold-standard GO terms, we compute AUPR and average F1 scores for each solution (Methods). On the GE datasets, solutions produced by DOMINO and NetBox showed the highest robustness over a wide range of subsampling fractions (Fig 6D and E). On the GWAS datasets, DOMINO's solutions scored highest (Fig EV2D and E).

A breakdown of the evaluation criteria by their properties is shown in Fig 6F.

Table 1 summarizes the benchmark results. DOMINO performed best on the GE datasets in five of the six criteria, and in all six criteria on the GWAS datasets. NetBox came second, performed best or timed for best in two criteria and second in the rest.

In addition, DOMINO ran much faster than the other algorithms, taking 1-3 orders of magnitude less time (Appendix Tables S4–S6). This speed allows to run DOMINO and the EMP procedure in reasonable time on a desktop machine. We also noticed that runtimes were markedly shorter on permuted datasets, probably since after permutation activity scores are spread more uniformly across the network, producing smaller modules.

### Analysis of large-scale networks

Our benchmark used the highly informative but relatively small DIP network (~3k nodes and ~ 5k edges) in order to allow systematic evaluation of multiple AMI methods on many datasets. Yet, much larger networks are currently available. To examine how DOMINO performs on larger network, we applied it on two state-of-the-art human networks: the HuRI network (8,272 nodes and 52,549 edges) (Luck et al, 2020) and STRING (with > 18K nodes and > 11M edges) (Szklarczyk et al, 2017). We also tested NetBox, the second-best performer in our benchmark, on these larger networks. The edges of the STRING network are weighted with a confidence score, ranging from 0 to 1000, based on the strength of their supporting evidence. To make the execution of the EMP feasible, we kept only edges with score > 900. The resulting network had 11,972 nodes and 243,385 edges. Setting a running time limit of 5 hrs, DOMINO

**Table 1.   Summary of the benchmark analysis.**

| Alg. | EHR | mEHR[a] | Robustness (F1)[b] | Robustness (AUPR)[b] | Biological Richness[c] | Intra-module homogeneity[c] |
|---|---|---|---|---|---|---|
| GE datasets | | | | | | |
| jAM_greedy | 0.052 ± 0.137 | 0.048 | 0.046 ± 0.117 | 0.062 ± 0.123 | 2.5 ± 9.837 | 0.325 ± 0.748 |
| jAM_SA | 0.236 ± 0.25 | 0.228 | 0.174 ± 0.188 | 0.204 ± 0.229 | 22 ± 18.02 | 1.335 ± 1.097 |
| Bionet | 0.398 ± 0.4 | 0.422 | 0.182 ± 0.15 | 0.321 ± 0.322 | 20 ± 23.819 | 1.929 ± 1.302 |
| NetBox | 0.719 ± 0.425 | 0.602 | 0.438 ± 0.266 | 0.632 ± 0.417 | 24 ± 32.301 | **2.575 ± 1.453** |
| KPM | 0.149 ± 0.296 | 0.177 | 0.13 ± 0.241 | 0.159 ± 0.297 | 8 ± 15.621 | 0.698 ± 0.983 |
| DOMINO | **0.891 ± 0.129** | **0.868** | **0.486 ± 0.192** | **0.776 ± 0.174** | **30.5 + 12.69** | 2.376 ± 0.754 |
| HotNet2 | 0.424 ± 0.429 | 0.378 | 0.183 ± 0.158 | 0.288 ± 0.23 | 9 ± 6.004 | 0.951 ± 0.791 |
| GWAS datasets | | | | | | |
| jAM_greedy | 0.151 ± 0.2 | 0.133 | 0.105 ± 0.138 | 0.125 ± 0.171 | 3.5 ± 15.481 | 1.165 ± 1.287 |
| jAM_SA | 0.053 ± 0.155 | 0.032 | 0.046 ± 0.134 | 0.058 ± 0.173 | 0 ± 11.83 | 0.327 ± 0.736 |
| Bionet | 0.298 ± 0.466 | 0.318 | 0.138 ± 0.21 | 0.267 ± 0.407 | 2 ± 8.634 | 0.857 ± 1.211 |
| NetBox | 0.694 ± 0.497 | 0.794 | 0.335 ± 0.298 | 0.617 ± 0.456 | **11 ± 8.485** | 1.172 ± 1.338 |
| KPM | 0.134 ± 0.312 | 0.148 | 0.144 ± 28 | 0.16 ± 0.324 | 0 ± 9.073 | 0.438 ± 0.719 |
| DOMINO | **0.844 ± 0.307** | **0.867** | **0.452 ± 0.268** | **0.673 ± 0.291** | **11 ± 9.55** | **2.085 ± 1.61** |
| HotNet2 | 0.081 ± 0.241 | 0.031 | 0.016 ± 0.037 | 0.061 ± 0.183 | 0 ± 0.843 | 0.036 ± 0.115 |

Per algorithm, average score over the ten datasets is shown. Best score in each criterion is in bold.
[a]Results are average over the top 10 modules.
[b]Results shown for subsampling fraction = 0.8.
[c]Results shown for Resnik cutoff = 3.

**Table 2.   Performance of DOMINO and NetBox on the larger networks.**

| Network | Alg. | EHR | mEHR[a] | Robustness (F1)[b] | Robustness (AUPR)[b] | Biological richness[c] | Intra-module homogeneity[c] |
|---|---|---|---|---|---|---|---|
| GE datasets | | | | | | | |
| HURI | NetBox | 0.505 ± 0.482 | 0.41 | 0.223 ± 0.228 | 0.458 ± 0.416 | **10 ± 25.738** | 1.084 ± 1.694 |
| | DOMINO | **0.881 ± 0.313** | **0.528** | **0.3 ± 0.289** | **0.642 ± 0.38** | 6.5 ± 17.515 | **1.354 ± 1.48** |
| STRING | NetBox | 0.18 ± 0.38 | 0.18 | 0.144 ± 0.305 | 0.177 ± 0.374 | 0 ± 39.314 | 0.477 ± 1.019 |
| | DOMINO | **0.939 ± 0.046** | **0.9** | **0.547 ± 0.282** | **0.788 ± 0.285** | **43 ± 43.818** | **2.326 ± 0.687** |
| GWAS datasets | | | | | | | |
| HURI | NetBox | 0.3 ± 0.483 | 0.28 | 0.08 ± 0.144 | 0.234 ± 0.399 | 0 ± 3.9 | 0.165 ± 0.371 |
| | DOMINO | **0.939 ± 0.585** | **0.419** | **0.547 ± 0.251** | **0.506 ± 0.447** | **4.5 ± 9.878** | **2.445 ± 0.533** |
| STRING | NetBox | 0.425 ± 0.438 | 0.367 | 0.328 ± 0.342 | 0.422 ± 0.438 | 12 ± 27.683 | 1.392 ± 1.371 |
| | DOMINO | **0.692 ± 0.371** | **0.782** | **0.389 ± 0.231** | **0.532 ± 0.346** | **13 ± 18.093** | 1.986 ± 1.629 |

Per algorithm, average score over the ten datasets is shown. Best score in each criterion is in bold.
[a]Results are average over the top 10 modules.
[b]Results shown for subsampling fraction = 0.8.
[c]Results shown for Resnik cutoff = 3.

completed all runs on both the HuRI and STRING networks, while NetBox did so on these two network for 8/10 and 2/10 of the GE datasets and for 9/10 and 9/10 of the GWAS datasets, respectively. Notably, DOMINO consistently outperformed NetBox on 23 of the 24 criteria on both networks and both types of datasets (Table 2). DOMINO also performed overall better when using different HG q-value thresholds (Table EV1). Taken together, these results demonstrate that DOMINO maintains high performance when applied to large networks as well.

### Analyzing the network contribution to non-specific GO enrichment bias

Understanding the causes for over-reporting of enriched GO terms is a key question that arises from our study. One prominent potential cause is the network topology, as the modules sought are connected subnetworks, and connectivity also reflects functional similarity. To explore the contribution of the network to the GO enrichment bias, we next detected modules in the underlying DIP network without

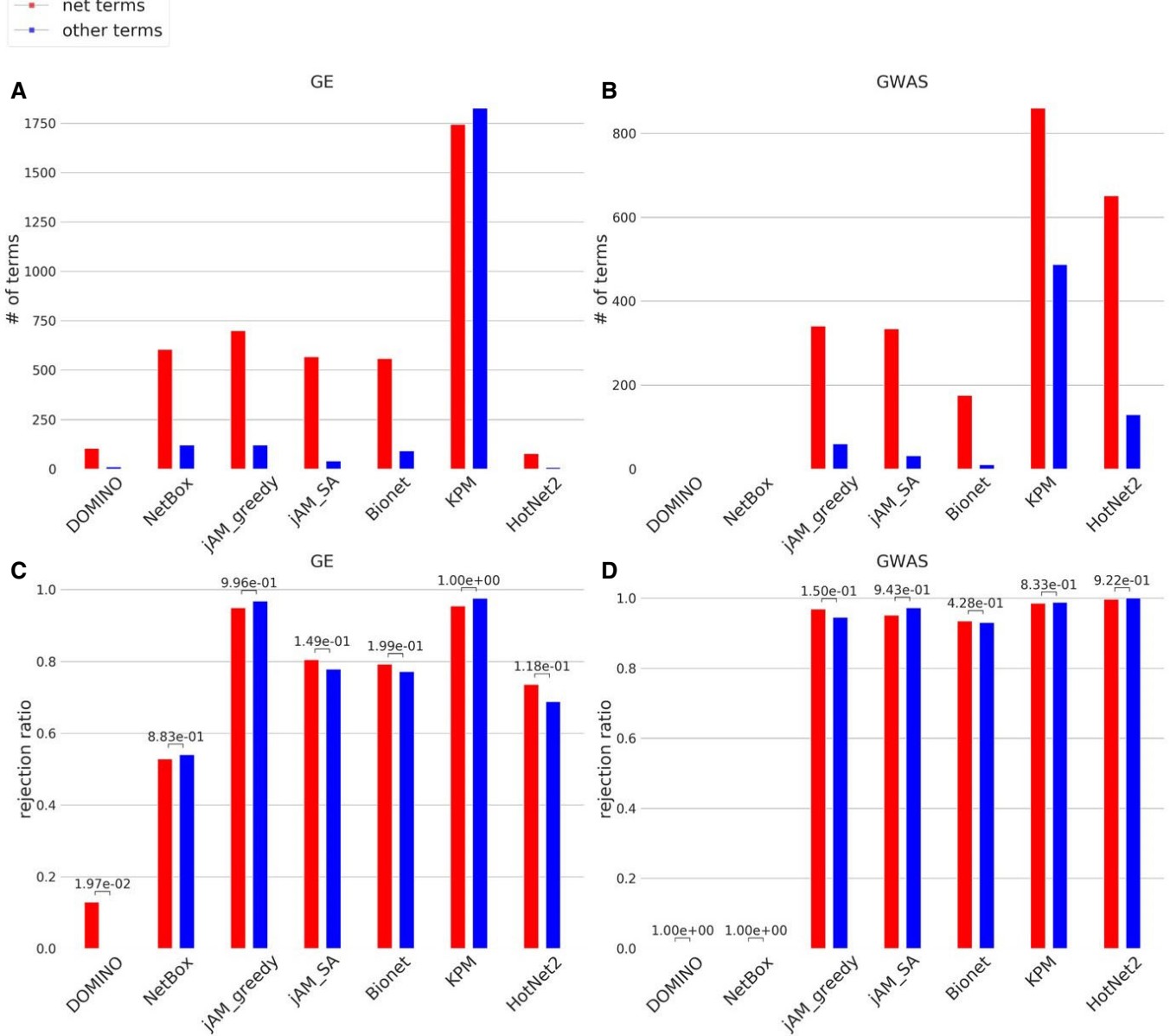

**Figure 7. Comparison of the GO terms identified by each benchmarked algorithm to the terms identified by using the network only (net-terms).**

A, B   Average number of net-terms and other terms. Only terms reported in four datasets or more were included. Note that no terms were reported in more than four GWAS datasets by DOMINO and NetBox, which obtained the best overall results (Table 1). (A) GE; (B) GWAS.

C, D   Average rejection ratio of net-terms and other terms. The rejection ratio of a GO term in an algorithm is the fraction of datasets in which the term appeared as significant but was not empirically validated (see Appendix). (C) GE; (D) GWAS. P-values were calculated by comparing the rejection ratios between net-terms and other terms using Mann–Whitney U one-sided test.

use of any condition-specific activity profile and identified the GO terms these modules were enriched for. Overall, 2,450 out of 6,573 (37%) BP GO terms were detected by this analysis, and we refer to them as *net-terms*. Notably, while net-terms were in general highly over-represented among the GO terms reported by AMI solutions (Fig 7A and B), these terms did not show higher rejection rate by the EMP procedure than the other BP GO terms (Fig 7C and D) (see Appendix for full details of this analysis). These results show that simple exclusion of GO net-terms from AMI analyses cannot replace

the empirical validation to lessen over-reporting of non-specific GO terms. Better understanding of the bias origin is required.

## Discussion

The fundamental task of active module identification (AMI) algorithms is to identify active modules in an underlying network based on context-specific gene activity profiles. The comparison of AMI

algorithms is challenging due to the complex nature of the solutions they produce. Algorithms differ markedly in the number, size, and properties of the modules they detect. Although AMI algorithms have been extensively used for almost two decades (Ideker *et al*, 2002), there is no accepted community benchmark for this task and no consensus on evaluation criteria. As active modules are often used to characterize the biological processes that are most relevant in the context of the profiled activity, we analyzed the solutions produced by AMI algorithms from the perspective of enrichment for GO terms annotating biological processes.

Previous works reported that the scheme used by the popular jActiveModule algorithm to score active modules is biased toward large modules and suggested ways to alleviate this bias (Nikolayeva *et al*, 2018) (preprint: Reyna *et al*, 2020). Our study reports on a different bias that is prevalent in AMI solutions: their tendency to report non-specific GO terms. Early on in our analysis, we observed that many enriched GO terms also appear on permuted datasets, suggesting that such enrichment stems from some proprieties of the network, algorithm, or the data that bias the results. To overcome this bias, we developed the EMP procedure, which empirically calibrates the enrichment scores and filters out non-specific terms. This procedure can be applied to any AMI algorithm.

To exemplify their merits, studies that present a novel AMI method usually report a collection of enriched gene sets (e.g., GO terms or pathways) obtained on the algorithm's solution and are biologically relevant to the analyzed condition. While this approach is valid for demonstrating capabilities of an algorithm, it is problematic for a systematic evaluation of algorithms, due lack of gold-standard bias-free set of biologically relevant GO terms for a given condition. An additional difficulty is the hierarchical structure of GO ontology. A previous benchmark of AMI algorithms used as an evaluation criterion the fold enrichment of the output genes using a single set of biologically relevant genes (He *et al*, 2017). In our work, we defined five novel evaluation criteria based on the GO terms enriched in a solution, each emphasizing a different aspect of the solution (Fig 6F).

We used these criteria to benchmark six popular AMI algorithms and DOMINO, a novel algorithm we developed, on ten GE and ten GWAS datasets, which collectively cover a very wide spectrum of biological conditions. Overall, DOMINO performed best, indicating its ability to produce "clean," stable, and concise modules. NetBox also scored high in our evaluation. Interestingly, both DOMINO and NetBox use binary gene activity scores. One may expect that binarizing measured activity scores could degrade relevant biological signals. However, at least on our benchmark, binarizing the data helped in reducing noise and detecting modules that are specifically relevant for the analyzed conditions. Further study of this observation is needed.

Notably, the algorithms that we tested substantially differ in their empirical validation rates. Some algorithms produced solutions with very low EHR (< 0.5), and therefore running the EMP on them was critical. While empirical correction is desirable and adds confidence to the reported results, it is computationally highly demanding even with a relatively small network such as DIP. Naturally, using larger networks makes this procedure even slower (Appendix Tables S4–S6). A notable advantage of DOMINO is the high validation rates: On our benchmark, its average EHR and mEHR were above 0.84, suggesting that DOMINO can be confidently run without empirical validation when computational resources are limited.

A common caveat in any report comparing a novel method to extant ones is that the new method may be better tuned to the data than the other methods. This may introduce a bias in the reported results. In our case, we could not tune each of the other AMI methods due to the long running time of EMP. Community efforts like the DREAM challenges (Choobdar *et al*, 2019) help reduce potential bias by allowing authors to calibrate their own methods on a common set of test datasets. To enable additional testing, the code of DOMINO, EMP, and the evaluation criteria is freely available at https://github.com/Shamir-Lab/.

In summary, in this study we (i) reported on a highly prevalent bias in popular AMI algorithms, which leads to non-specific calls of enriched GO terms, (ii) developed a procedure to allow for the correction of this bias, (iii) introduced novel criteria for evaluation of AMI solutions, and (iv) developed DOMINO—a novel AMI algorithm with low rate of non-specific calls and better performance across most of the criteria.

# Materials and Methods

### The Louvain algorithm in DOMINO

The Louvain algorithm is a fast community detection method for large network (Blondel *et al*, 2008). This method aims to optimize an objective function by iteratively moving nodes between community to improve the objective function and fusing together the nodes of each community. In our benchmark, we used a variant (Lambiotte *et al*, 2008) that incorporates a resolution parameter denoted *r*, which we set to 0.15.

### Threshold for testing relevant slices

Slices that contain only a few active nodes are unlikely to be relevant. Testing multiple such slices would diminish the significance of genuine relevant slices. Therefore, we test for relevance only slices that satisfy either.

$$\frac{\#active\ nodes\ in\ slice}{\#active\ nodes\ in\ network} \geq 0.1.$$

or

$$\frac{\#active\ nodes\ in\ slice}{\#\ nodes\ in\ slice} \geq \alpha.$$

where

$$\alpha = \min\left(0.7, \frac{\#active\ nodes\ in\ network}{\#\ nodes\ in\ network}\right) * \left(1 + \frac{100}{\sqrt{\#\ nodes\ in\ network}}\right).$$

### The PCST application in DOMINO

In PCST (Johnson *et al*, 2000), nodes have values called prizes, and edges have values called penalties. All values are non-negative. The goal is to find a subtree *T* that maximizes the sum of the prizes of nodes in *T* minus the sum penalties of the edges in it, i.e., $\sum_{v \in T} p(v) - \sum_{e \in T} c(e)$ where $p(v)$ is the prize of node $v$, and $c(e)$ is the cost of edge $e$.

The node prizes are computed by diffusing the activity of the nodes using influence propagation with the linear threshold model

(Kempe *et al*, 2015). The process is iterative: Initially, the set of active nodes is as defined by the input. In each iteration, an inactive node is activated if the sum of the influence of its active neighbors exceeds θ = 0.5. The influence of a node that has $k$ neighbors on each neighbor is $\frac{1}{k}$. Activated nodes remain so in all subsequent iterations. The process ends when no new node is activated. If v became active in iteration $l$ then $p(v) = \beta^l$ where $\beta = \max\left(0, 1 - 3 * \frac{\#active\ nodes\ in\ network}{\#nodes\ in\ network}\right)$. We define the penalty of edge $e$ as $c(e) = 0$ if it is connected to an active node, and $c(e) = 1 - \varepsilon$ otherwise (we used $\varepsilon = 10^{-4}$). PCST is NP-hard but good heuristics are available. In DOMINO, we used FAST-PCST (Hegde *et al*, 2014). The resulting subgraph obtained by solving PCST on each slice is called its sub-slice. See Fig 3C.

### The Newman–Girvan algorithm in DOMINO

The Newman–Girvan (NG) algorithm is a community detection method (Girvan & Newman, 2002). This method iteratively removes edges using the Betweenness-centrality metric for edges and recomputes the modularity score for each intermediate graph. Let $M_i$ be the modularity score for the graph in iteration $i$. The process continues until a stopping criterion is met. The stopping criterion we used in step (2b) is $\frac{\log(\#\ of\ nodes\ in\ sub-slice)}{\log(\#\ of\ nodes\ in\ network)} \leq M_i$.

### Derivation of *P*-values and q-values for the GE and GWAS datasets

For the GE datasets, we calculated *P*-values for differential expression between test and control conditions using edgeR (Robinson *et al*, 2010) for RNA-seq and Student's t-test for microarray datasets. We computed q-values using Benjamini–Hochberg FDR method (Benjamini & Hochberg, 1995). For GWAS, we used SNP-level *P*-values for association with the analyzed trait to derive gene-level association *P*-values using PASCAL (Lamparter *et al*, 2016), using the sum chi-square option and flanks of 50k bps around genes. We computed q-values using Benjamini–Hochberg FDR method (Benjamini & Hochberg, 1995).

### Criteria for evaluating AMI solutions

We defined five novel criteria to allow systematic evaluation of solutions provided by AMI algorithms. For a specific solution, we considered the list of BP GO terms that passed the HG enrichment test (HG terms) and the terms that passed the EMP validation procedure (EV terms).

### Solution-level criteria

#### *Empirical-to-Hypergeometric Ratio (EHR)*
We define the *Empirical-to-Hypergeometric Ratio* (EHR) as the ratio between the number of EV terms and reported HG terms. EHR summarizes the tendency of an algorithm to report non-specific GO terms, with values close to 1.0 indicating good solutions while values close to 0 indicating poor ones. EHR reflects the precision (true-positive rate) of a solution.

#### *Biological richness*
This criterion quantifies the biological information collectively captured by the solution's EV terms. As there is high redundancy among GO terms—mainly due to the hierarchical structure of the

GO ontology—we use the method implemented in REVIGO (Supek *et al*, 2011) to derive a non-redundant set of EV terms. This method is based on a similarity matrix of GO terms, which is generated using Resnik similarity score (Resnik, 1999). The *biological richness score* is defined as the number of non-redundant EV terms in a solution. We calculated this measure using different similarity cutoffs (1.0 to 4.0 in REVIGO).

#### *Solution robustness*
This criterion evaluates the robustness of a solution to incomplete gene activity data. It compares the EV terms obtained on the original dataset with those obtained on randomly subsampled datasets, where non-sampled gene levels are treated as missing. We repeated this procedure for subsampling fractions 0.6, 0.7, 0.8, and 0.9, iterating each fraction 100 times. Using the EV terms of the full dataset as the positive set, we computed average precision, recall and F1 scores across these iterations. Another perspective is provided by the examination of the frequency by which GO terms are detected in the subsampled datasets: higher frequency for a specific EV term implies higher robustness. We measured this robustness aspect of a solution using AUPR, in which EV terms are ranked according to their frequency across iterations (again, using EV terms detected on the full dataset as the positive instances). Note that cases in which an algorithm results in many empty solutions (that is, solutions with no enriched GO terms) and a few non-empty ones that are enriched for true EV terms can yield a high but misleading AUPR score. Therefore, we validated that the fraction of non-empty solutions obtained by the algorithms on the subsampled runs is high: All the algorithms achieved around 60% or more non-empty solutions on GE data (Appendix Fig S4).

### Module-level criteria

#### *Module-Level EHR (mEHR)*
This criterion calculates a single module's EHR. We define the module-level EHR (*mEHR*), as the ratio between the number of a module's EV terms and HG terms (Appendix Fig S3). We score each solution by averaging the mEHR of its $k$ top-ranked modules ($k$ values ranging from 1 to 20).

#### *Intra-module homogeneity*
This index measures the homogeneity of the biological signal that is captured by each module compared to the biological signal in the entire solution. For its calculation, we build a (complete) graph for the solution's EV terms (GO graph) in which nodes represent the EV terms and the weights on the edges are the pairwise Resnik similarity score (Appendix Fig S3B). Next, edges whose weight is below a cutoff are removed. The *intra-module homogeneity* is defined as the module's relative edge-density in this GO graph:

$$\frac{\left(\frac{of\ edges\ in\ module's\ GO\ graph}{\#\ of\ edges\ in\ a\ complete\ graph\ of\ that\ size}\right)}{\left(\frac{\#\ of\ edges\ in\ the\ solution's\ GO\ graph}{\#\ of\ edges\ in\ a\ complete\ graph\ of\ the\ same\ size}\right)}$$

We calculate the intra-module homogeneity score for a solution by averaging its modules' scores (Appendix Fig S3B). We repeat this test for a range of similarity cutoffs—from 1.0 to 4.0. This criterion provides a complementary view on top of the one captured by the

biological richness criterion, by characterizing the biological coherence of the reported modules.

## Data availability statement

The code for DOMINO, EMP, and the benchmark criteria is available at https://github.com/Shamir-Lab. The datasets used in this study are listed in the Appendix Tables S2 and S3. The GE datasets and the gene scores of both GE and GWAS datasets are also available at https://github.com/Shamir-Lab/EMP under the "datasets" folder.

**Expanded View** for this article is available online.

## Acknowledgements

We thank the reviewers for insightful comments which helped us to substantially improve the paper. This study was supported in part by German-Israeli Project DFG RE 4193/1-1 (to RS and RE), by the Israel Science Foundation grants No. 1339/18 (to RS) and No. 2118/19 (to RE), by Len Blavatnik And The Blavatnik Family Foundation (to RS) and the Koret-UC Berkeley-Tel Aviv University Initiative in Computational Biology and Bioinformatics (to R.E.). HL was supported in part by a fellowship from the Edmond J. Safra Center for Bioinformatics at Tel Aviv University. R.E. is a Faculty Fellow of the Edmond J. Safra Center for Bioinformatics at Tel Aviv University.

## Author contributions

RE and RS conceived the project. HL designed and developed the algorithms, implemented them, and performed the analysis under the supervision of RE and RS. All authors wrote the manuscript and approved the final manuscript.

## Conflict of interest

The authors declare that they have no conflict of interest.

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
