## [Review Process File · Molecular Systems Biology]

DOMINO: a network-based active module identification algorithm with reduced rate of false calls

Hagai Levi, Ran Elkon, and Ron Shamir
DOI: [10.15252/msb.20209593](https://doi.org/10.15252/msb.20209593)

Corresponding author(s): Ron Shamir (rshamir@tau.ac.il) , Ran Elkon (rael@tauex.tau.ac.il)

Review Timeline:

Submission Date:	24th Mar 20
Editorial Decision:	18th Jun 20
Revision Received:	18th Oct 20
Editorial Decision:	3rd Nov 20
Revision Received:	9th Nov 20
Accepted:	11th Nov 20

Editor: Jingyi Hou

Transaction Report:

Thank you again for submitting your work to Molecular Systems Biology. We have now heard back from the three reviewers who agreed to evaluate your study. As you will see below, the reviewers acknowledge that the presented findings and method seem potentially interesting. They raise however a series of concerns, which we would ask you to address in a major revision.

I think that the reviewers' recommendations are rather clear and there is no need to reiterate the comments listed below. In particular, reviewer #1 is concerned about the relatively small size of the DIP networks that are used for the comparison between NBMD methods, and this reviewer provides constructive suggestions in that regard.

All issues raised by the reviewers need to be satisfactorily addressed. As you may already know, our editorial policy allows in principle a single round of major revision and it is therefore essential to provide responses to the reviewers' comments that are as complete as possible. Please feel free to contact me in case you would like to discuss in further detail any of the issues raised by the reviewers.

On a more editorial level, we would ask you to address the following issues.

REFEREE REPORTS

Reviewer #1:

This study presents a rigorous comparison between 6 network-based module detection (NBMD) schemes, and a new NBMD scheme, named DOMINO, which it shows to outperform other methods. It starts by showing a problem in NBMD methods, wherein many significant GO terms are identified in modules enriched for active genes but also in permuted datasets. The correction it offers is an empirical pipeline, i.e., to create a score distribution per GO term, and compare the observed versus expected score per GO term. Given this pipeline, the study describes several measures for NBMD evaluation. The Domino algorithm is a nice combination of existing methods, such as a community detection method for dissecting the graph into modules and a prize-collecting Steiner tree for identifying subnetworks consisting of active nodes.

Major:

The study has several shortcomings: It presents a problem that was potentially inflated, an empirical NBMD method that integrates previously published methods and thus has limited novelty, and applies NBMD methods and particularly DOMINO to a PPI network that is very small relative to

today's networks (3,000 nodes compared to ~20,000, and 5,000 interactions compared to ~400,000), which puts a question mark on DOMINO's applicability.

The introduction is not well written for several reasons. When writing about modules, the definition of a module should be clear, esp. given that this is a computational study. Moreover, modules should be described in more depth (e.g., consider including the seminal paper 'From molecular to modular cell biology' Nature, 1999). The methods for module evaluation should also be described thoroughly.

The introduction struck me as a clutter of citations. The first and second paragraphs of the introduction cite studies almost arbitrarily. The sentence "With the ever-increasing availability of genomic, transcriptomic and proteomic data" is followed by references to studies from 2008, 2018 and 2011. Why are these specific studies cited? Are they seminal in any sense, which I doubt? If going back a decade made sense, why cite a paper from 2018 that discusses a combination of genetic interactions with PPI (genetic interactions not a particular demonstration of a genomic, transcriptomic, or a proteomic assay but rather a viability assay), especially when integrative approaches are described later in the text.

The introduction continues with a reference to a review of networks from 2006. I recommend either pointing to the first studies that used networks, or to an up-to-date review of networks. Another examples: "In these networks, each node represents a cellular subunit (e.g. a protein) and each edge represents a relationship between two subunits", which cites a cytoscape app, the DIP and the String database. These are tools and not conceptual papers, namely where the concept emerged. If you point to a tool then refer to it as such. Similarly, if you point to a review it would be helpful to add "reviewed in"..

Another example of poor choice of citations: "This ability of NBMD methods is especially critical for the functional interpretation of Genome-Wide Association Studies (GWASs) (Visscher et al, 2012). " - why cite a review of '5 years of GWAS discovery' 8 years after it was published?

This frustrating style of writing and citing goes on and on in the introduction, which should be revised throughout. Such arbitrary and context-less citing is not acceptable.

Regarding module definition. Modules are first defined in the text as highly scoring subnetworks: "The core of the NBMD task is to pinpoint highly scoring sets of interacting nodes, where the score of each node (i.e. the activity score) is derived from the data ..". In the next sentence, modules are defined differently "An NBMD solution is composed of a set of modules that are enriched for the activity signal.". So, it is highly scoring or is it enriched? Is every high-scoring gene there? Or can some module genes not be high scoring? What about modules that are based on dense interactions and not activity?

The text continues with inaccurate references to GO. "wherein the GO (The Gene Ontology Consortium, 2019) functions" However, GO provides gene annotation, not functions. GO does have a 'molecular function' ontology, but it seems that the authors actually refer to the GO 'biological process' ontology. They mention GO biological process annotations in some of the evaluation criteria, and Fig. 5B presents biological process terms. The introduction continuously refers to 'Functional GO category', which should be revised as it is misleading and simply inaccurate. It should be made clear in the introduction and throughout which type of GO terms were used in the different analyses.

Another example for inaccurate writing: "The most popular approach for biologically interpreting a gene set is the Hypergeometric (HG) test, where the proportion of genes annotated for a certain property (GO functional category) in the set is compared to a background set of genes." The popular approach is to use GO annotations and enrichment. Hypergeometric (HG) test is just the

statistical test, which could be a different one, see for example Gorilla and GSEA statistical tests.

The introduction points to several NBMD studies as particularly successful "NBMD methods are among the most effective for fulfilling this task (Marbach et al, 2016; Barrenas et al, 2009; Cowen et al, 2017)" however the study does not include them in the comparison. This requires justification and some discussion.

The entire comparison between NBMD methods relies on DIP, which, as the authors write, is very small compared to today's PPI networks. Huang et al found that "STRING, ConsensusPathDB, and GIANT networks having the best performance overall... Correcting for size, we find that the DIP network provides the highest efficiency (value per interaction)." However, today's applications rarely use DIP. Thus, at least some tests should be done with current networks, esp. for assessing DOMINO relative to another method.

The GO enrichment analysis is at the heart of the comparison. How was the GO enrichment calculated? Specifically, what was the background list for the enrichment analysis? Which GO evidence codes were used? What is the threshold of the corrected p-value used in Fig. 1A? Were only GO biological process terms considered throughout, or the entire GO?

In any case, to make the results unbiased by the small DIP network, the background list used to assess GO enrichment should be the list of proteins that appear as nodes in the DIP network. Was this done?

Upon analyzing GO term enrichment, corrected p-value thresholds that are considered significant in real world bioinformatic applications are lower than 0.05, since 0.05 is too permissive. For example, the GOrilla tool uses thresholds that start at 10^{-3} and go down to 10^{-10} . Thus, the analysis should consider only GO terms with corrected p-value of 10^{-3} or lower.

Gene annotations to be included per gene should not be based on certain GO evidence codes, including Inferred from Physical Interaction (IPI), Inferred from Mutant Phenotype (IMP), Inferred from Genetic Interaction (IGI), and Inferred from Expression Pattern (IEP), as these bias the analysis which aims to identify PPI modules based on gene expression patterns and mutation patterns. For all the reasons I mentioned, it seems that the GO enrichment analysis was too permissive, which resulted in identifying many GO terms that falsely appear as significant. This potentially lead to inflation of GO terms and created a problem that, if properly analyzed, might not be that frequent.

How is permutation achieved? Is it uniform, or are activity scores switched between genes with the same connectivity? This should be described and discussed.

Thresholds used in the algorithms seem arbitrary (e.g., 0.2 in the community detection, etc). Please elaborate on why those thresholds and not others.

To check the applicability of the Domino algorithm to up-to-date PPI network, apply it to the human interactome of BioGRID or STRING, and evaluate results. This would also test the suitability of the different thresholds used throughout.

Add run times on DIP and on the up-to-date PPI network.

The discussion is totally focused on results without giving a broader view. There is not a single reference to another study.

The different evaluation criteria that the study suggests are thoughtful. If this is a novelty of the

study, then a more elaborate discussion of measures used by other NBMD studies is required.

Page 14, 15 and 17 contains a figure without a legend. The fig. seems identical.

Minor:

Introduction: "distinct biological endpoints" - endpoint is unclear.

"Unexpectedly, our analysis revealed that algorithms often obtained modules enriched for a high number of GO terms even when run on permuted datasets" remove 'Unexpectedly,' because this is not so unexpected.

Table S4: numbers should appear as simple as possible. E.g., 4.60E-01 should appear as 0.46. 0.00E+00 should be 0.

Figure 1B is not intuitive. Consider using another type of plot.

In the end, the question is whether a certain module is enriched. However in the empirical pipeline, fig 2 panel C, the maximal score per GO term is used. Why this choice, and not the score of each GO term per module?

Fig. 4: A boxplot is better than a histogram; the median is more representative of the results than mean. It seems that the ratio is the number of experimentally validated terms out of the total terms detected. How can the ratio be greater than 1?

Fig. 5: show results also for the GWAS (this should be in all relevant figures).

Fig. 5B justifies the EMP, and thus can come before Domino in the text. Also, Fig. 6A is mentioned in the text before 5B.

Fig. 7 is not helpful.

Reviewer #2:

In this manuscript, the authors have performed a comparative analysis of multiple algorithms to detect active networks in genomics data, propose new metrics for benchmarking such algorithms, and define a new method to address shortcomings of the existing tools. The paper is clearly written and the comparative analysis across multiple methods is interesting. I have three major concerns regarding the analysis.

First, active network methods project experimental results on a reference gene/protein interaction network. In this work, DIP was used, which seems appropriate to me. However, I wonder how many of the outcomes of the paper are dependent on the input network, and how that input network may alter results differently depending on the algorithm. Are these results robust to changes in DIP or, more broadly, the use of different input networks?

Second, I wonder if there is a more straightforward way to address false positives in enrichment analysis of permuted data. While I agree that permutation analysis is a sensible way to assess false positive rates, I wonder if the false positives may be address directly by altering method parameters rather than carrying out the slow, ad hoc EMP analysis. For example, what background gene sets

were used in the hypergeometric test? Seeing the same GO annotations appear in permuted data suggests a bias in that background. As noted above, I worry that these outcomes may also be biased by the topology of the input network. For these reasons I'm not convinced that EMP is a necessary or desirable way to clean the results, despite its technical validity.

Third, the DOMINO algorithm appears to be designed to generate relatively small modules. Do the DOMINO modules contain fewer genes? How does modules size affect each of the scoring metrics? Would higher significance thresholding in other methods improve their performance across these metrics?

Other comments:

It is not clear why "biological richness" is a general scoring criterion. I think the diversity of enriched annotations would depend on the data set being analyzed.

I'm not sure Fig 1B is necessary. I did not find it to be visually intuitive, and I wonder if it contributes more than the Venn diagrams in Fig 1A.

Text in Figs 1A and 4 is too small.

Reviewer #3:

The paper "DOMINO: a novel network-based module detection algorithm with reduced rate of false calls" by Levi et al. starts with the observation that network-based module discovery (NBMD) algorithms frequently produce "active" modules with significant overlap with GO terms not only when using real omics data, such as gene expression or GWAS data, but also for permuted data. The authors propose an empirical procedure (EMP) to adjust for this, and develop five performance criteria to better evaluate NMMD algorithms. Finally they propose their own new method, DOMINO, that (largely) outperforms six established methods on these criteria.

The paper is sound and relevant. It is also well written for the most part (see my minor comments on some potential improvements that could still be made).

Major comments:

I like the structure of the paper, first reporting the observation of a potential bias (significant modules by GO enrichment for permuted data), then coming up with a way to overcome this (EMP) and a battery of evaluation criteria, and only proposing a new algorithm in the end. The paper would have substantial merit even without the last part, i.e. DOMINO, and in a way it may not be ideal to have these two aspects in the same work. This is because it is always problematic when a new method is evaluated against the state-of-the-art with criteria set up by the method inventors. Unsurprisingly, the new method always comes out with great performance. This is in part, because the inventors could design (and sometimes tune) their own method to perform well under these criteria, which introduces bias. Maybe more problematic, at least when the criteria are sound, as in the present case, is that the other methods are usually run with default parameters and often do not receive the same attention to adapt them as the new method. This is of course a general issue which applies to many method papers, yet it is worth mentioning that efforts like the DREAM challenges address exactly this bias by organising analysis challenges for which all participants

have the chance to adapt their method in a learning phase and then compete with equal chances. While I do not suggest to change the paper fundamentally, I do think it would be useful to refer to this bias, and make it very easy to apply the benchmarking performed in this paper to any other method, i.e. provide the code not only for DOMINO, but also for the benchmarking, in case that's not already on the git (as the datasets).

In their discussion the authors write "An additional future task is to understand better the sources of the bias that causes over-reporting of enriched GO terms.". My hypothesis is that modules with high GO enrichment for permutation data likely correspond to subnetworks (i.e. network modules) that have a very high overlap with some GO terms, such that any omics data, including randomized versions, that select subsets of such network modules still come out significant. In other words, the network structure may be the main driver of the GO enrichment. I think this hypothesis could (and should) easily be tested. In fact methods that just partition networks into modules have recently been systematically evaluated in a DREAM challenge - not by using GO enrichment but GWAS enrichment as an evaluation method, see PMID: 31471613. One could for example take some top methods of this competition (or at least the Newman-Girvan modularity detection algorithm used in step 0 of DOMINO) to identify such GO terms, and check if the problem persists when removing or flagging them.

Minor comments

There's a typo in the abstract: " evlautating NBMD"

"we designed a method that evaluates the empirical significance of GO terms" : I'd add "by permutation analysis"

"gene activity scores" scores is a bit of a misnomer for the GWAS scores, since, in contrast to gene expression data, genetic associations infer little information about activity itself (active genes may not carry any genetic variants that modulate their activity in which case they won't be detected by GWAS.) Maybe 'gene relevance scores' would be sufficiently generic a term.

Used terminology like node "activity", "slices" or "dissected" could be replaced with wording used more commonly in network science, like for example node attribute, partition, communities, etc.

Fig. 1B: It took me a while to (hopefully) understand what is shown. (It didn't help that there are 10 datasets and 10 permutations, maybe increase the latter to resolve this.) If the output for each algorithm and data type (GE/GWAS) consists just of 10 numbers, I believe a barplot (with 10 bars rather than color coding) or a color-coded table would do a better job. Also in this case no sorting is needed and one could have explicit reference to each of the 10 datasets.

"For some two decades": almost/more than?

Would be good to report the sample size for the GWAS studies. The height paper is very early and more recent studies (see GWAS catalog or the aforementioned DREAM challenge for summary stats) will have much more power.

It might be helpful for the reader if the authors would explain in more clarity how their particular clustering problem fits into the more general literature. It seems the underlying problem is known in computer/network science as community detection with node attributes (usually with discrete attributes, but continuous attributes have been considered as well in the literature). A more

detailed discussion of the benefits / drawbacks of taking additional node information into account would also be beneficial.

How would the performance of DOMINO change if the Newman-Girvan modularity detection algorithm used in step 0 would be replaced by other algorithms designed to detect network communities (again the DREAM challenge I mentioned tried to identify the most efficient ones for PPI and other gene-gene networks)?

Reviewer #1:

The study has several shortcomings: It presents a problem that was potentially inflated, an empirical NBMD method that integrates previously published methods and thus has limited novelty, and applies NBMD methods and particularly DOMINO to a PPI network that is very small relative to today's networks (3,000 nodes compared to ~20,000, and 5,000 interactions compared to ~400,000), which puts a question mark on DOMINO's applicability.

Response: We revised DOMINO to enable running it on larger networks. We applied DOMINO and NetBox with the EMP procedure on the larger HuRI and STRING networks in order to validate GO terms. The results show that DOMINO had consistent performance advantage in all networks. Further analyses that we performed show that the problem presented is not inflated and persists on large networks as well. See additional response to specific comments below.

Regarding the "limited novelty" of DOMINO, we respectfully and strongly disagree. Most novel methods use ideas from previously published research. It is the combination of these multiple ideas that matters. The improved results of DOMINO in a field of research that has been studied for two decades speak for themselves.

The introduction is not well written for several reasons. When writing about modules, the definition of a module should be clear, esp. given that this is a computational study. Moreover, modules should be described in more depth (e.g., consider including the seminal paper 'From molecular to modular cell biology' Nature, 1999). The methods for module evaluation should also be described thoroughly.

The introduction struck me as a clutter of citations. The first and second paragraphs of the introduction cite studies almost arbitrarily. The sentence "With the ever-increasing availability of genomic, transcriptomic and proteomic data" is followed by references to studies from 2008, 2018 and 2011. Why are these specific studies cited? Are they seminal in any sense, which I doubt? If going back a decade made sense, why cite a paper from 2018 that discusses a combination of genetic interactions with PPI (genetic interactions not a particular demonstration of a genomic, transcriptomic, or a proteomic assay but rather a viability assay), especially when integrative approaches are described later in the text.

The introduction continues with a reference to a review of networks from 2006. I recommend either pointing to the first studies that used networks, or to an up-to-date review of networks. Another examples: "In these networks, each node represents a cellular subunit (e.g. a protein) and each edge represents a relationship between two subunits", which cites a cytoscape app, the DIP and the String database. These are tools and not conceptual papers, namely where the concept emerged. If you point to a tool then refer to it as such. Similarly, if you point to a review it would be helpful to add "reviewed in".. Another example of poor choice of citations: "This ability of NBMD methods is especially critical for the functional interpretation of Genome-Wide Association Studies (GWASs) (Visscher et al, 2012). " - why cite a review of '5 years of GWAS discovery' 8 years after it

was published?

This frustrating style of writing and citing goes on and on in the introduction, which should be revised throughout. Such arbitrary and context-less citing is not acceptable.

Response: We agree with the criticism. We completely rewrote the introduction, and, as suggested, substantially changed the citations.

Regarding module definition. Modules are first defined in the text as highly scoring subnetworks: "The core of the NBMD task is to pinpoint highly scoring sets of interacting nodes, where the score of each node (i.e. the activity score) is derived from the data". In the next sentence, modules are defined differently "An NBMD solution is composed of a set of modules that are enriched for the activity signal.". So, it is highly scoring or is it enriched? Is every high-scoring gene there? Or can some module genes not be high scoring? What about modules that are based on dense interactions and not activity?

Response: We rephrased the text and clarified module definitions (Page 2). In response to a comment of reviewer 3 we also changed the terminology: We now use "active module identification" (AMI) and not NBMD, which may be confused with network-only module discovery. We also changed the paper's title to reflect this.

The text continues with inaccurate references to GO. "wherein the GO (The Gene Ontology Consortium, 2019) functions" However, GO provides gene annotation, not functions. GO does have a 'molecular function' ontology, but it seems that the authors actually refer to the GO 'biological process' ontology. They mention GO biological process annotations in some of the evaluation criteria, and Fig. 5B presents biological process terms. The introduction continuously refers to 'Functional GO category', which should be revised as it is misleading and simply inaccurate. It should be made clear in the introduction and throughout which type of GO terms were used in the different analyses.

Response: Indeed, in our analyses, GO terms of the biological process (BP) ontology were used. We now clarify this in the text (page 5) and give exact details on the particular GO terms we tested for enrichment.

Another example for inaccurate writing: "The most popular approach for biologically interpreting a gene set is the Hypergeometric (HG) test, where the proportion of genes annotated for a certain property (GO functional category) in the set is compared to a background set of genes. " The popular approach is to use GO annotations and enrichment. Hypergeometric (HG) test is just the statistical test, which could be a different one, see for example Gorilla and GSEA statistical tests.

Response: This section was removed from the Introduction.

The introduction points to several NBMD studies as particularly successful "NBMD methods are among the most effective for fulfilling this task (Marbach et al, 2016; Barrenas et al, 2009; Cowen et al, 2017)" however the study does not include them in the comparison. This requires justification and some discussion.

Response: We rephrased this paragraph and changed the citations. AMI methods used in the cited studies are either included our benchmark or share similar ideas with algorithms in our benchmark (Nakka et al, 2016 use HotNet2; Chang et al 2015 solves a version of PCST, as do Bionet and DOMINO in the benchmark, Fernandez-Tajes et al, 2019 is based on ideas from jActiveModules greedy).

The entire comparison between NBMD methods relies on DIP, which, as the authors write, is very small compared to today's PPI networks. Huang et al found that "STRING, ConsensusPathDB, and GIANT networks having the best performance overall... Correcting for size, we find that the DIP network provides the highest efficiency (value per interaction)." However, today's applications rarely use DIP. Thus, at least some tests should be done with current networks, esp. for assessing DOMINO relative to another method.

Response: We added comparison of DOMINO to NetBox on two other larger networks: HuRI (8272 nodes, 52,549 edges) network and STRING (11972 nodes, 243,385 edges with confidence larger than 900). We now report DOMINO scored best overall in all three networks.

The GO enrichment analysis is at the heart of the comparison. How was the GO enrichment calculated? Specifically, what was the background list for the enrichment analysis? Which GO evidence codes were used? What is the threshold of the corrected p-value used in Fig. 1A? Were only GO biological process terms considered throughout, or the entire GO?

In any case, to make the results unbiased by the small DIP network, the background list used to assess GO enrichment should be the list of proteins that appear as nodes in the DIP network. Was this done?

Response:

- The background list for the enrichment analysis indeed comprises only genes that appears in the network.
 - Following the reviewer's comment, to avoid potential biases in our analyses, we now excluded the following code evidence: IPI, IMP, IGI, IEP, HMP, HGI, HEP
 - We corrected for multiple testing (BH-FDR) and took terms with $p\text{-value} < 0.05$
 - We considered only GO biological process terms in our analysis
- All these details are mentioned in the manuscript (pages 5 and 7)

Upon analyzing GO term enrichment, corrected p-value thresholds that are considered significant in real world bioinformatic applications are lower than 0.05, since 0.05 is too permissive. For example, the GOrilla tool uses thresholds that start at 10^{-3} and go down to 10^{-10} . Thus, the analysis should consider only GO terms with corrected p-value of 10^{-3} or lower.

Response: We rerun our benchmark with different p-value thresholds: 0.05, 0.01, 0.001. (see External Table 1). While DOMINO was less dominant for some thresholds, it was still the best performing algorithm overall. We also argue that the need for thresholds lower than 10^{-3} used in previous studies was partially due to issues we addressed in our paper.

Gene annotations to be included per gene should not be based on certain GO evidence codes, including Inferred from Physical Interaction (IPI), Inferred from Mutant Phenotype (IMP), Inferred from Genetic Interaction (IGI), and Inferred from Expression Pattern (IEP), as these bias the analysis which aims to identify PPI modules based on gene expression patterns and mutation patterns.

Response: We agree with this important point and filtered out the following evidence codes: IPI, IMP, IGI, IEP, HMP, HGI, HEP

For all the reasons I mentioned, it seems that the GO enrichment analysis was too permissive, which resulted in identifying many GO terms that falsely appear as significant.

This potentially leads to inflation of GO terms and created a problem that, if properly analyzed, might not be that frequent.

Response: We addressed the factors mentioned by the reviewer, and the problem has persisted.

How is permutation achieved? Is it uniform, or are activity scores switched between genes with the same connectivity? This should be described and discussed.

Response: We shuffled the scores uniformly at random, without considering the network structure. This is explained in page 5.

Thresholds used in the algorithms seem arbitrary (e.g., 0.2 in the community detection, etc). Please elaborate on why those thresholds and not others.

Response: We reached these thresholds based on empirical testing. We mention this issue in page 19

To check the applicability of the Domino algorithm to up-to-date PPI network, apply it to the human interactome of BioGRID or STRING, and evaluate results. This would also test the suitability of the different thresholds used throughout.

Response: We conducted analysis on two other popular networks: HuRI and STRING. See "*Analysis of large-scale networks*" in the Results section.

Add run times on DIP and on the up-to-date PPI network.

Response: We added runtimes. See Tables S4-6 and page 15. Notably, the new version of DOMINO is now 1-3 orders of magnitude faster than the other tested methods.

The discussion is totally focused on results without giving a broader view. There is not a single reference to another study.

Response: We revised Discussion section, referring also to related works.

The different evaluation criteria that the study suggests are thoughtful. If this is a novelty of the study, then a more elaborate discussion of measures used by other NBMD studies is required.

Response: Yes, the criteria are novel. We now discuss them and a criterion used by another benchmark on page 18

Page 14, 15 and 17 contains a figure without a legend. The fig. seems identical.

Response: fixed

Minor:

Introduction: "distinct biological endpoints" - endpoint is unclear.

Response: we revised this phrase as part of the introduction revision

"Unexpectedly, our analysis revealed that algorithms often obtained

modules enriched for a high number of GO terms even when run on permuted datasets" remove 'Unexpectedly,' because this is not so unexpected.

Response: We now say "Remarkably" instead. We still feel it is surprising, since reporting meaningless/non-specific results that show up also on random data is definitely not what any of those algorithms intended.

Table S4: numbers should appear as simple as possible. E.g., 4.60E-01 should appear as 0.46. 0.00E+00 should be 0.

Response: Fixed. Note that the old Table S4 is now integrated into Table 1.

Figure 1B is not intuitive. Consider using another type of plot.

Response: We changed the figure. According to Reviewer #3's suggestion, we also increased # of permutations to 100.

In the end, the question is whether a certain module is enriched. However in the empirical pipeline, fig 2 panel C, the maximal score per GO term is used. Why this choice, and not the score of each GO term per module?

Response: First, the goal of the EMP procedure is to examine if the GO terms detected on an AMI solution are specific to the context of the analyzed biological condition. Thus, we take the maximal score per GO term and compare it to the Null distribution. This allows us to reject non-specific terms. Second, the Null distribution is generated on random permutations. Each AMI algorithm run on such permuted profile generates a solution with a different number of modules of different sizes. This does not allow building Null distribution per module. Thirdly, the mEHR provides an evaluation score per module. The higher this score, the higher the confidence the signal captured by the module is indeed specific to the analyzed condition.

Fig. 4: A boxplot is better than a histogram; the median is more representative of the results than mean.

Response: We did include the median (red cross) to Figure 4

It seems that the ratio is the number of experimentally validated terms out of the total terms detected. How can the ratio be greater than 1?

Response: The EHR indeed cannot be greater than 1. As the results for individual datasets (the grey dots in Fig 4 A, B) show, it never is.

Fig. 5: show results also for the GWAS (this should be in all relevant figures).

Response: The results for the GWAS are shown in supplementary figures (S3, S4).

Fig. 5B justifies the EMP, and thus can come before Domino in the text. Also, Fig. 6A is mentioned in the text before 5B.

Response: Since this figure also includes DOMINO's results, we chose to place it after introducing DOMINO. Fig. 5B is now referred to before Fig 6A.

Fig. 7 is not helpful.

Response: We now include it as a subfigure of Figure 6, so it is less prominent. We feel it helps to understand the different characteristics of each criterion and their relationship.

Reviewer #2:

In this manuscript, the authors have performed a comparative analysis of multiple algorithms to detect active networks in genomics data, propose new metrics for benchmarking such algorithms, and define a new method to address shortcomings of the existing tools. The paper is clearly written and the comparative analysis across multiple methods is interesting. I have three major concerns regarding the analysis.

Response: We thank the reviewer for the positive opinion

First, active network methods project experimental results on a reference gene/protein interaction network. In this work, DIP was used, which seems appropriate to me. However, I wonder how many of the outcomes of the paper are dependent on the input network, and how that input network may alter results differently depending on the algorithm. Are these results robust to changes in DIP or, more broadly, the use of different input networks?

Response: We now report on results for the DIP, HuRI and STRING networks (page 15; Table 2). Overall, the results on other networks were consistent with the results on our original network.

Second, I wonder if there is a more straightforward way to address false positives in enrichment analysis of permuted data. While I agree that permutation analysis is a sensible way to assess false positive rates, I wonder if the false positives may be address directly by altering method parameters rather than carrying out the slow, ad hoc EMP analysis.

Response: Tuning parameters of each algorithm was impractical for our compute-intensive benchmark. We now added an extensive analysis of an alternative way for detecting false positive terms based on the potential bias introduced by network topology (Results section: Analyzing the network contribution to non-specific GO enrichment bias and Appendix, pages 30-31). We also provide results for different q-value thresholds (External Table 1). These results show that there is no simple alternative for the EMP procedure.

For example, what background gene sets were used in the hypergeometric test? Seeing the same GO annotations appear in permuted data suggests a bias in that background.

Response: The background set of the HG test comprises the genes that are included in the network. This is mentioned in page 5.

As noted above, I worry that these outcomes may also be biased by the topology of the input network. For these reasons I'm not convinced that EMP is a necessary or desirable way to clean the results, despite its technical validity.

Response: As mentioned above, we added an extensive analysis of an alternative way for detecting false positive terms based on the bias of the network (page 17 and the Appendix). Our analysis indicates that additional factors are involved in the bias and simple correction based on network topology is insufficient.

Third, the DOMINO algorithm appears to be designed to generate relatively small modules. Do the DOMINO modules contain fewer genes? How does modules size affect each of the scoring metrics? Would higher significance thresholding in other methods improve their performance across these metrics?

Response: We do provide module size statistics. See Figure S1, Figure S2. One can see DOMINO's modules are not substantially smaller than those of other algorithms (with the exception of jAM SA).

Other comments:

It is not clear why "biological richness" is a general scoring criterion. I think the diversity of enriched annotations would depend on the data set being analyzed.

Response: We agree that the context of the assay needs to be taken into account when interpreting the results of each criterion, and emphasize this on page 12. Yet, if diverse biological processes were empirically validated by EMP, it indicates that the AMI solution successfully captured multiple molecular facets that are specifically relevant to the analyzed biological condition (while AMI solution with less rich diversity missed some components).

I'm not sure Fig 1B is necessary. I did not find it to be visually intuitive, and I wonder if it contributes more than the Venn diagrams in Fig 1A.

Response: We revised Fig. 1B and made it visually simpler.

Text in Figs 1A and 4 is too small.

Response: Fixed

Reviewer #3:

The paper "DOMINO: a novel network-based module detection algorithm with reduced rate of false calls" by Levi et al. starts with the observation that network-based module discovery (NBMD) algorithms frequently produce "active" modules with significant overlap with GO terms not only when using real omics data, such as gene expression or GWAS data, but also for permuted data. The authors propose an empirical procedure (EMP) to adjust for this, and develop five performance criteria to better evaluate NMMD algorithms. Finally they propose their own new method, DOMINO, that (largely) outperforms six established methods on these criteria.

The paper is sound and relevant. It is also well written for the most part (see my minor comments on some potential improvements that could still be made).

Response: We thank the reviewer for the positive evaluation.

Major comments:

I like the structure of the paper, first reporting the observation of a potential bias (significant modules by GO enrichment for permuted data), then coming up with a way to overcome this (EMP) and a battery of evaluation criteria, and only proposing a new algorithm in the end. The paper would have substantial merit even without the last part, i.e. DOMINO, and in a way it may not be ideal to have these two aspects in the same work. This is because it is always problematic when a new method is evaluated against the state-of-the-art with criteria set up by the method inventors. Unsurprisingly, the new method always comes out with great performance. This is in part, because the inventors could design (and sometimes tune) their own method to perform well under these criteria, which introduces bias. Maybe more problematic, at least when the criteria are sound, as in the present case, is that the other methods are usually run with default parameters and often do not receive the same attention to adapt them as the new method. This is of course a general issue which applies to many method papers, yet it is worth mentioning that efforts like the DREAM challenges address exactly this bias by organising analysis challenges for which all participants have the chance to adapt their method in a learning phase and then compete with equal chances. While I do not suggest to change the paper fundamentally, I do think it would be useful to refer to this bias, and make it very easy to apply the benchmarking performed in this paper to any other method, i.e. provide the code not only for DOMINO, but also for the benchmarking, in case that's not already on the git (as the datasets).

Response: We agree with the concerns and now discuss them (page 19). We now provide in Github the code for the evaluation criteria and the datasets used in our benchmark. We did not include the other methods code as this would necessitate substantial additional technical work (and authors permissions).

In their discussion the authors write "An additional future task is to understand better the sources of the bias that causes over-reporting of enriched GO terms.". My hypothesis is that modules with high GO enrichment for permutation data likely correspond to subnetworks (i.e. network modules) that have a very high overlap with some GO terms, such that any omics data, including randomized versions, that select subsets of such network modules still come out significant. In other words, the network structure may be the main driver of the GO enrichment. I think this hypothesis could (and should) easily be tested. In fact methods that just partition networks into modules have recently been systematically evaluated in a DREAM challenge - not by using GO enrichment but GWAS enrichment as an evaluation method, see PMID: 31471613. One could for example take some top methods of this competition (or at least the Newman-Girvan modularity detection algorithm used in step 0 of DOMINO) to identify such GO terms, and check if the problem persists when removing or flagging them.

Response: We thank the reviewer for this thoughtful suggestion. We conducted additional analysis that compares GO terms in modules discovered by AMI methods to those discovered by a community detection method, which discover modules based on the network only. We took the reviewer's suggestion and used the M1 method, one of the top performers in the DREAM competition.

Briefly, we extracted a list of GO terms using M1. We called these terms net-terms. We tested the overlap between net-terms and the terms that were discovered by AMI algorithms. We found that while net-terms were in general highly over-represented among

the GO terms reported by AMI solutions, these terms did not show significantly higher rejection rate by the EMP procedure than the other BP GO terms. This analysis is described in Page 17 and the Appendix). We concluded that while a substantial part of the rejected terms were net-terms, the association between the two types of terms was mild, and many identified terms that are not net-terms must be rejected as well. Overall, one cannot use net-terms as an alternative to EMP for filtering GO terms.

Minor comments

There's a typo in the abstract: " evlautating NBMD"

Response: Fixed.

"we designed a method that evaluates the empirical significance of GO terms" : I'd add "by permutation analysis"

Response: we revised this sentence according the reviewer's comment.

"gene activity scores" scores is a bit of a misnomer for the GWAS scores, since, in contrast to gene expression data, genetic associations infer little information about activity itself (active genes may not carry any genetic variants that modulate their activity in which case they won't be detected by GWAS.) Maybe 'gene relevance scores' would be sufficiently generic a term.

Response: We prefer to use the commonly used term activity scores. We added a comment about GWAS terminology in page 2.

Used terminology like node "activity", "slices" or "dissected" could be replaced with wording used more commonly in network science, like for example node attribute, partition, communities, etc.

Response: We replaced "dissect" by "partition". We kept "slices", as they refer to specific subsets in our algorithm.

Fig. 1B: It took me a while to (hopefully) understand what is shown. (It didn't help that there are 10 datasets and 10 permutations, maybe increase the latter to resolve this.) If the output for each algorithm and data type (GE/GWAS) consists just of 10 numbers, I believe a barplot (with 10 bars rather than color coding) or a color-coded table would do a better job. Also in this case no sorting is needed and one could have explicit reference to each of the 10 datasets.

Response: We changed this figure to make it more visually intuitive, and additionally, as suggested, increased the number of permutations to 100.

"For some two decades": almost/more than?

Response: The right word is "almost". Fixed.

Would be good to report the sample size for the GWAS studies. The height paper is very early and more recent studies (see GWAS catalog or the aforementioned DREAM challenge for summary stats) will have much more power.

Response: Sample sizes are reported now in Table S3. We kept the early height paper data in order to avoid an expensive rerun of the benchmark.

It might be helpful for the reader if the authors would explain in more clarity how their particular clustering problem fits into the more general literature. It seems the underlying problem is known in computer/network science as community detection with node attributes (usually with discrete attributes, but continuous attributes have been considered as well in the literature). A more detailed discussion of the benefits / drawbacks of taking additional node information into account would also be beneficial.

Response: Our revised Introduction section now explains and discusses most of these issues.

How would the performance of DOMINO change if the Newman-Girvan modularity detection algorithm used in step 0 would be replaced by other algorithms designed to detect network communities (again the DREAM challenge I mentioned tried to identify the most efficient ones for PPI and other gene-gene networks)?

Response: We thank the reviewer for this helpful idea. We replaced the Newman-Girvan algorithm by the Louvain algorithm in step 0. This led to substantial speed-up of DOMINO which allowed its application to larger networks. As we show, overall, the revised algorithm performed as good as the original one on the DIP network.

Thank you for sending us your revised manuscript. We have now heard back from the two reviewers who were asked to evaluate your study. As you will see below the reviewers are satisfied with the modifications made and think that the study is now suitable for publication.

Before we can formally accept your manuscript, we would ask you to address a few remaining editorial issues listed below.

REFEREE REPORTS

Reviewer #1:

My comments were addressed.

Reviewer #3:

The authors sufficiently addressed my comments.

The Authors have addressed all editorial concerns.

ACCEPTED**11th Nov 2020**

Thank you again for sending us your revised manuscript. We are now satisfied with the modifications made and I am pleased to inform you that your paper has been accepted for publication.

Corresponding Author Name: Ron Shamir, Ran Elkon

Journal Submitted to: Molecular System Biology

Manuscript Number: MSB-20-9593